# PARG-deficient tumor cells have an increased dependence on EXO1/FEN1-mediated DNA repair

Christina Andronikou [1,2,3,4], Kamila Burdova[5], Diego Dibitetto[1,4], Cor Lieftink[6,7], Elke Malzer[6,7], Hendrik J Kuiken [6,7], Ewa Gogola[2], Arnab Ray Chaudhuri[8], Roderick L Beijersbergen[6,7], Hana Hanzlikova[1,5], Jos Jonkers [2,3✉] & Sven Rottenberg [1,2,4✉]

## Abstract

Targeting poly(ADP-ribose) glycohydrolase (PARG) is currently explored as a therapeutic approach to treat various cancer types, but we have a poor understanding of the specific genetic vulnerabilities that would make cancer cells susceptible to such a tailored therapy. Moreover, the identification of such vulnerabilities is of interest for targeting BRCA2;p53-deficient tumors that have acquired resistance to poly(ADP-ribose) polymerase inhibitors (PARPi) through loss of PARG expression. Here, by performing whole-genome CRISPR/Cas9 drop-out screens, we identify various genes involved in DNA repair to be essential for the survival of PARG;BRCA2;p53-deficient cells. In particular, our findings reveal EXO1 and FEN1 as major synthetic lethal interactors of PARG loss. We provide evidence for compromised replication fork progression, DNA single-strand break repair, and Okazaki fragment processing in PARG;BRCA2;p53-deficient cells, alterations that exacerbate the effects of EXO1/FEN1 inhibition and become lethal in this context. Since this sensitivity is dependent on BRCA2 defects, we propose to target EXO1/FEN1 in PARPi-resistant tumors that have lost PARG activity. Moreover, EXO1/FEN1 targeting may be a useful strategy for enhancing the effect of PARG inhibitors in homologous recombination-deficient tumors.

**Keywords** DNA Repair; PARG; BRCA2; EXO1; FEN1
**Subject Categories** Cancer; DNA Replication, Recombination & Repair; Post-translational Modifications & Proteolysis

## Introduction

Poly(ADP-ribosyl)ation (PARylation) is a dynamic posttranslational modification, consisting of chains of ADP-ribose repeats that play a critical role in the regulation of a plethora of cellular processes (Leung, 2014). PARylation has been described to mediate the repair of DNA single-stranded or double-stranded breaks (SSBs or DSBs), the stabilization of replication forks (RF), regulation of chromatin remodeling, and the processing of unligated Okazaki fragments (Ray Chaudhuri and Nussenzweig, 2017; Hanzlikova et al, 2018). Members of the poly(ADP-ribose)polymerase (PARP) superfamily of proteins catalyze such PAR chain formation, with PARP1 being the main enzyme among them (Amé et al, 2004). PARP1 contains a highly conserved (ADP-ribosyl)-transferase (ART) domain, which catalyzes the cleavage of nicotinamide adenine nucleotide ($NAD^+$) into nicotinamide and ADP-ribose molecules, which are then being transferred on PARP1 itself and other proteins forming branched or linear chains of PARylation (Kim et al, 2005). PARylation chains need to be subsequently removed through hydrolysis of the glycosidic bonds between the ADP-ribose units, which are then recycled as ATP precursors for the regeneration of $NAD^+$ (Murata et al, 2019). Poly(ADP-ribose)glycohydrolase (PARG) is the main ADP-ribose hydrolase carrying out this de-PARylation step. It is specialized in the endo- or exo-glycohydrolation of PAR chains, leaving the last mono-ADP-ribose units attached to serve as a substrate for other erasers, such as the ADP-ribose hydrolase 3 (ARH3) and the terminal ADP-ribose protein glycohydrolase (TARG1) (Pascal and Ellenberger, 2015; Oka et al, 2006; Hanzlikova et al, 2020; Fontana et al, 2017; Sharifi et al, 2013).

In recent years, PARP and PARG inhibition has been explored to treat cancers with high levels of replication stress and genomic instability due to defective DNA repair (Slade, 2020). PARP and PARG inhibitors exacerbate such tumor vulnerabilities, resulting in increased DNA damage and the accumulation of unresolved replication intermediates that cause replication and mitotic catastrophe (Slade, 2020). In particular, the important role that PARP1 plays in the regulation of the DNA damage response (DDR) has been exploited therapeutically for cancer (Farmer et al, 2005; Bryant et al, 2005). Upon the occurrence of a DNA single-strand break (SSB), PARP1 is immediately recruited to the DNA lesion on the chromatin, where it PARylates itself and other proteins, which acts as a signal for the recruitment of DNA damage response (DDR) proteins (Tallis et al, 2014). Four PARP inhibitors (PARPi) that are blocking the activity of PARP1/2 and are trapping PARP1 on the chromatin have been

[1]Institute of Animal Pathology, Vetsuisse Faculty, University of Bern, 3012 Bern, Switzerland. [2]Division of Molecular Pathology, The Netherlands Cancer Institute, 1066CX Amsterdam, The Netherlands. [3]Oncode Institute, Amsterdam, The Netherlands. [4]Cancer Therapy Resistance Cluster and Bern Center for Precision Medicine, Department for Biomedical Research, University of Bern, 3088 Bern, Switzerland. [5]Laboratory of Genome Dynamics, Institute of Molecular Genetics of the Czech Academy of Sciences, 142 20 Prague 4, Czech Republic. [6]Division of Molecular Carcinogenesis, The Netherlands Cancer Institute, 1066CX Amsterdam, The Netherlands. [7]The Netherlands Cancer Institute Robotics and Screening Center, The Netherlands Cancer Institute, 1066CX Amsterdam, The Netherlands. [8]Department of Molecular Genetics, Erasmus MC Cancer Institute, Erasmus University Medical Center, 3015GD Rotterdam, The Netherlands. ✉E-mail: j.jonkers@nki.nl; sven.rottenberg@unibe.ch

clinically approved for the treatment of breast (olaparib, talazoparib), ovarian (olaparib, niraparib, rucaparib), prostate (rucaparib), and pancreatic (olaparib) cancer (Kim and Nam, 2022). PARP1/2 inhibition gives rise to DNA double-strand breaks (DSBs), as a result of (1) SSBs that cannot be efficiently repaired and are converted into DSBs during replication and (2) collapsed replication forks after encountering trapped PARP1 (Murai et al, 2012, 2014; Pommier et al, 2016). Since the repair of these lesions requires an intact homologous recombination (HR) pathway, HR-deficient cancers (e.g., due to dysfunctional BRCA1/2) amass chromosomal aberrations resulting in cell death by mitotic catastrophe after PARPi treatment (Lupo and Trusolino, 2014; Pommier et al, 2016; Rose et al, 2020). Despite the successful introduction of PARPi in the clinic, patients are eventually developing tumors that are resistant to these treatments, and various mechanisms have been put forward to explain this phenomenon (Gogola et al, 2017; D'Andrea, 2018; Dias et al, 2021; Baxter et al, 2022). Among these mechanisms, the restoration of HR caused by re-established BRCA1/2 function has been detected in human cancer (Edwards et al, 2008; Sakai et al, 2008; ter Brugge et al, 2016; Barber et al, 2013; Domchek, 2017; Lin et al, 2019). Nevertheless, secondary BRCA1/2 mutations explain only some of the PARPi cases (Ang et al, 2013; Tobalina et al, 2021; Baxter et al, 2022). In our recent work, we have found that PARG loss is the most frequently detected PARPi-resistance mechanism in mammary tumors from BRCA2-deficient triple-negative breast cancer (TNBC) mouse models (Gogola et al, 2018; Bhin et al, 2023). Mechanistically, we uncovered that PARG loss mediates PARPi resistance by partially restoring PARP1 signaling, and we observed that biopsies of TNBC and high serous ovarian carcinoma patients carry a substantial percentage of tumor cells with low expression of PARG and increased PARylation (Gogola et al, 2018). This indicates that these cells may be selected out by the PARPi treatment and contribute to PARPi resistance. Hence, understanding the genetic vulnerabilities of PARG- and BRCA2-deficient cells may be useful to develop novel therapeutic strategies to target PARPi-resistant tumors that have not genetically reversed BRCA2 function.

In addition to PARP, PARG inhibition has also been explored for cancer treatment, following the resolution of the structure of the PARG catalytic site (Slade et al, 2011; Dunstan et al, 2012; Kim et al, 2012; Barkauskaite et al, 2013). This led to the development of several PARGi, including PDD00017272/3 (PDDX-004/PDDX-001), JA2131 and COH34 (James et al, 2016; Houl et al, 2019; Chen and Yu, 2019). Complementary to the use of PARPi for targeting HR deficiencies, PARG inhibitors seem to be useful to exploit deficiencies in the DNA replication machinery of cancer cells (Pillay et al, 2019; Houl et al, 2019). However, little is known about the specific genetic vulnerabilities of cells that are PARGi-sensitive. We believe that this knowledge is crucial to further optimize the use of PARGi in precision oncology.

To chart the landscape of genes that become essential when PARG function is impaired, we performed two complementary CRISPR-Cas9 whole-genome drop-out screen approaches, applying both genetic perturbation and chemical PARG inhibition. Using our BRCA2;p53-deficient mammary tumor cells (KB2P), we found EXO1 as a common hit in both screens and FEN1 among the top hits in the genetic screen. Following their validation, we show that PARG;BRCA2-deficient cells carry increased replication defects which are amplified and result in cell death when EXO1 and FEN1 are inhibited. Moreover, we demonstrate that PARG-deficient cells exhibit a defect in the repair of PAR-regulated lesions, while

additional FEN1/EXO1 depletion leads to the accumulation of unresolved ssDNA gaps, which can serve as a source of toxicity for HR-deficient cells. Together, our findings strongly suggest FEN1 and EXO1 to be useful as pharmaceutical targets for PARG-deficient tumor cells, as well as for the optimization of PARGi in precision oncology.

# Results

## CRISPR/Cas9 drop-out screens reveal DNA repair-associated genes to be essential for the survival of PARG/BRCA2-deficient cells

To identify vulnerabilities of PARG deficiency in BRCA2-deficient tumors, we first set out to generate PARG-deficient $Brca2^{-/-};Trp53^{-/-}$ mammary tumor cells that recapitulate the previously described phenotype of PARPi resistance (Gogola et al, 2018). For this purpose, we genetically depleted $Parg$ in the $Brca2^{-/-};Trp53^{-/-}$ mammary tumor cell line KB2P1.21, derived from our $K14cre;Trp53^{F/F};Brca2^{F/F}$ mouse model for $BRCA2$-mutated breast cancer (Jonkers et al, 2001; Evers et al, 2008). Two separate approaches were applied for CRISPR/Cas9-mediated targeting of $Parg$: (1) A single sgRNA targeting exon 3 (sgRNA3-1), resulting in a clonal cell line with a homozygous $Parg$ frameshift mutation (KB2P-P1) (Fig. EV1A); (2) Introduction of a large $Parg$ deletion by simultaneous targeting of exons 3 (sgRNA3-2) and 9 (sgRNA9), resulting in a clonal KB2P-P2 cell line with the homozygous depletion of catalytically active PARG isoforms (O'Sullivan et al, 2019) (Fig. EV1A). For the generation of the KB2P-P1 clone, both the Cas9 and sgRNA sequences were introduced with independent lentiviral transductions, whereas for the KB2P-P2, we used ribonucleoprotein (RNP) transfer of the Cas9:sgRNA complexes. The newly generated clonal cell lines KB2P-P1 & P2 recapitulated the previously observed phenotypes: resistance to the PARPi olaparib (Fig. EV1B) and increased PARylation levels following treatment with the SSB-inducing alkylating agent methyl methanesulfonate (MMS) (Fig. EV1C).

Subsequently, we introduced the genome-wide YUSA mouse CRISPR/Cas9 library v2 into the KB2P-P2 cells as well as the isogenic PARG-proficient KB2P control cells. The YUSA library contains 90,230 sgRNA sequences, targeting in total 18,424 genes in the mouse genome (Tzelepis et al, 2016). This unbiased approach allowed us to detect genes that have a synthetic lethal (SL) interaction with the loss of PARG. After lentiviral transduction of the YUSA library and subsequent selection with puromycin for 3 days, cells were collected as reference (day 0) or propagated for an additional 8 days (day 8) (Fig. 1A). Next, the genomic DNA was isolated, and the sgRNA construct-specific sequences were amplified. Analysis of the amplified sequences derived from the cells harvested on day 0 and day 8 of three biological screen replicates revealed the specific and reproducible drop-out of sgRNAs targeting 221 different genes in the PARG-deficient cells (Fig. 1B). STRING analysis of these 221 hits revealed five main clusters of genes based on their molecular functionality (Fig. EV1D). In cluster 1 mismatch repair (MMR), DNA replication, nucleotide excision repair (NER), and base excision repair (BER) were the pathways with the highest representation in the network (Figs. 1C and EV1E). These four KEGG pathways were represented by 14 genes from the total candidate list. The gene that scored the

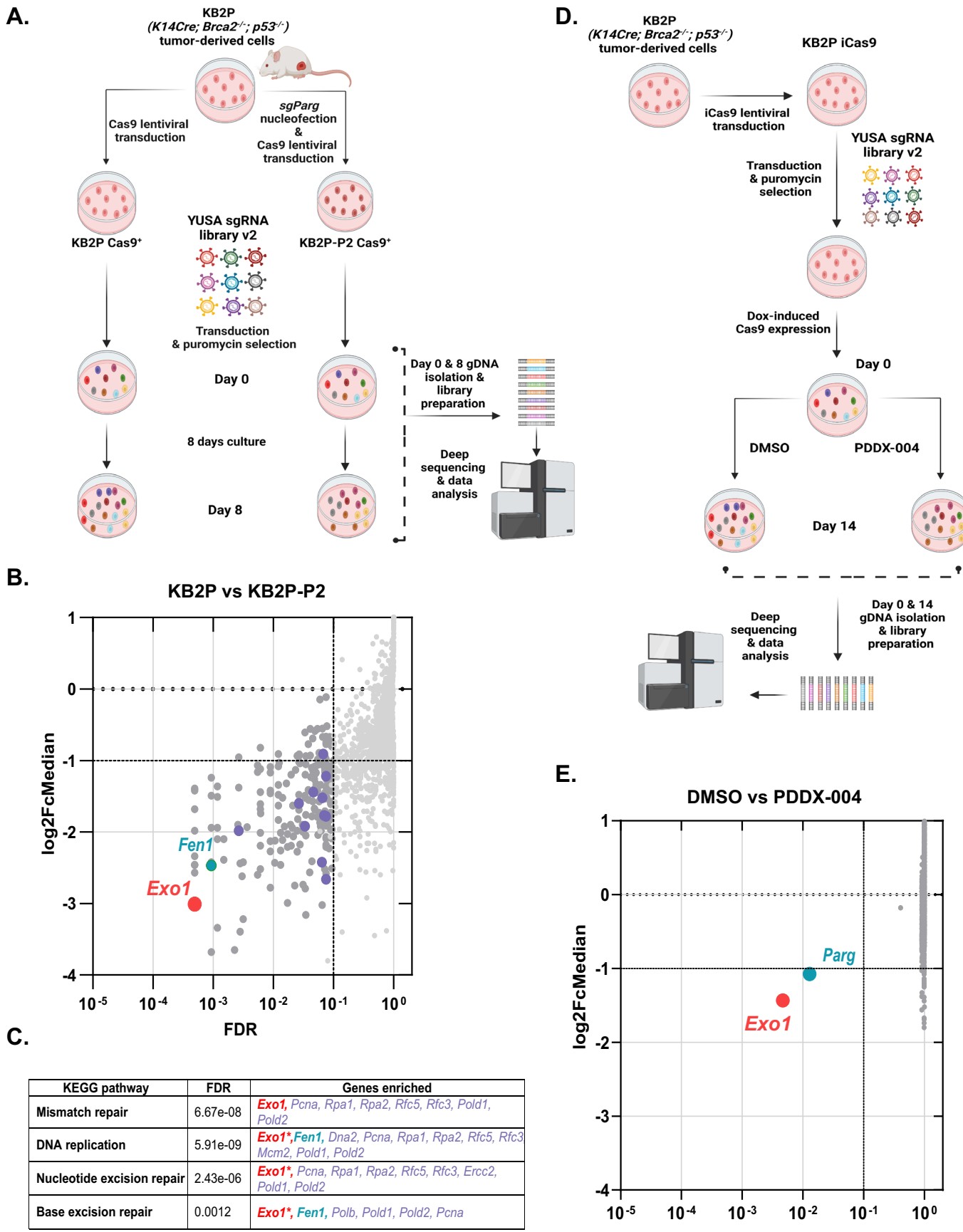

**C.**

| KEGG pathway | FDR | Genes enriched |
|---|---|---|
| Mismatch repair | 6.67e-08 | *Exo1, Pcna, Rpa1, Rpa2, Rfc5, Rfc3, Pold1, Pold2* |
| DNA replication | 5.91e-09 | *Exo1*,Fen1, Dna2, Pcna, Rpa1, Rpa2, Rfc5, Rfc3 Mcm2, Pold1, Pold2* |
| Nucleotide excision repair | 2.43e-06 | *Exo1*, Pcna, Rpa1, Rpa2, Rfc5, Rfc3, Ercc2, Pold1, Pold2* |
| Base excision repair | 0.0012 | *Exo1*, Fen1, Polb, Pold1, Pold2, Pcna* |

◀ **Figure 1.  CRISPR/Cas9 drop-out screens reveal essential genes for the survival of PARG-deficient *Brca2⁻/⁻;p53⁻/⁻* mammary tumor cells.**

(A) Outline of the CRISPR/Cas9 drop-out screen in PARG-proficient KB2P and PARG-deficient KB2P-P2 cells. (B) Dot plot representing the log2 fold-change ratio versus the false-discovery rate (FDR) of the comparison of KB2P versus KB2P-P2 cells and highlighting the hits with FDR < = 0.1. To eliminate artifacts, the genes that were already depleted at the seeding day of screen (day 0) were removed and additional quantile normalization was performed. Data derived from three biological replicates were analyzed using MAGeCK. Colored dots indicate the genes of Cluster 1 from the STRING analysis shown in Fig. EV1D. (C) Table annotating the most enriched molecular pathways (KEGG database) and associated genes according to the STRING analysis of the 221 genes that scored in the screen. *Exo1** indicates the participation of EXO1 in the annotated pathway, based on literature (Keijzers et al, 2015). (D) Outline of the CRISPR/Cas9 drop-out screen in KB2P cells cultured in DMSO or with 100 nM of the PARG inhibitor PDDX-004. (E) Dot plot representing the log2 fold-change ratio versus the FDR of the comparison of DMSO versus PDDX-004 treated KB2P cells and highlighting the hits with FDR < = 0.1. To eliminate artifacts, the genes that were already depleted on the seeding day of screen (day 0) were removed. Data derived from two biological replicates were analyzed using MAGeCK. Source data are available online for this figure.

highest in our genetic screen and which is reported to contribute to all of the four pathways is *Exo1*, which encodes a 5' to 3' exonuclease (Keijzers et al, 2015) (Fig. 1B,C).

We next sought to independently validate these results using a genome-wide drop-out screen approach, in which we inhibited PARG chemically (Fig. 1D). For this screen, we introduced an inducible Cas9 system into the KB2P cells. After the lentiviral transduction with the YUSA library and subsequent selection with puromycin, Cas9 expression was induced with doxycycline for 2 days, and cells were collected as reference (day 0) or propagated for 15 days, either in a DMSO-control condition or in the presence of the PARG inhibitor PDDX-004 (James et al, 2016; Waszkowycz et al, 2018), which effectively inhibits mouse PARG (Gogola et al, 2018; Bhin et al, 2023). For the screen, we applied 100 nM of PDDX-004, a concentration that conferred olaparib resistance and MMS-induced increase in PARylation levels of KB2P cells (Fig. EV1F,G). Strikingly, independent sgRNAs targeting *Exo1* were also depleted in this screen, and *Exo1* scored as a top hit (Fig. 1E). However, in addition to *Exo1*, sgRNAs targeting *Parg* were also depleted, indicating a PDDX inhibitor toxicity for PARG-deficient KB2P cells. (Fig. 1E). Indeed, both KB2P-P1 and –P2 cells were highly sensitive to treatment with the PDDX-004 inhibitor (Fig. EV2A), paralleled by a similar response to the structurally similar compound, PDDX-001 (Fig. EV2B), as evidenced in long-term viability assays. In contrast, the PARG-proficient KB2P-NT cells displayed resilience to these treatments. Corroborating these viability data, PDDX-004 treatment led to increased DNA damage specifically in PARG-deficient KB2P-P2 cells but not PARG-proficient KB2P-NT cells. This was evidenced by a notable accumulation of γH2AX foci under these treatment conditions (Fig. EV2C). To discern whether the cytotoxicity of PDDX compounds involves a PARP1 trapping mechanism akin to that observed with PARP inhibitors, we employed the trapping assay as described by Murai et al, 2012. Contrary to expectations, PDDX-004 treatment did not lead to an increase in the chromatin-bound PARP1 in KB2P-NT and KB2P-P2 cells after induction of DNA damage with MMS. Instead, it rather led to an additional increase in the ADP-ribosylation levels of KB2P-P2 cells (Fig. EV2D). Subsequently, we set out to investigate in more detail how PDDX-004 treatment modulates ADP-ribosylation levels of KB2P-NT and KB2P-P2 cells under unperturbed conditions. Using high-content indirect immunofluorescence with PCNA staining to determine G1 (2n, PCNA−), S phase (PCNA + ), and G2 (4n, PCNA−) cells, we observed that PDDX-004 treatment not only significantly increased ADP-ribosylation (PAR/MAR) levels in KB2P-NT cells (two-way ANOVA; $P < 0.0001$) but also further augmented the already heightened PAR/MAR levels of KB2P-P2 cells (two-way ANOVA; $P = 0.0010$) (Fig. EV2E). Given that KB2P-P2 clonal cells lack all catalytically active isoforms due to their large *Parg* deletion (Fig. EV1A), these results hint

at the possibility that their increased sensitivity to PDDX may stem from off-target inhibition of other ADP-ribose hydrolases.

Despite the potential for off-target effects of the PDDX inhibitor, *Exo1* emerged as a significant candidate in our screens for genetic depletion and chemical inhibition of PARG, suggesting its critical role for the survival of BRCA2-deficient cells under PARG loss.

### *Exo1* loss is synthetic lethal to *Parg* depletion in BRCA2-deficient but not BRCA2-proficient cells

To validate *Exo1* as an essential gene, we used CRISPR/Cas9-mediated targeting of *Exo1* in KB2P cells using two different sgRNAs. This resulted in a mixed cell population composed of cells with wild-type (wt) *Exo1* alleles and cells containing alleles with frameshift mutations. These mixed cell populations were then used in a competition assay, in which cells were treated with increasing concentrations of the two independent PARGi, PDDX-004 and PDDX-001 (Figs. 2A and EV2F). In contrast to the vehicle-treated control, we found a significant decrease in the fraction of mutated *Exo1* alleles (two-tailed Student's *t* test, $P = 0.0338$ and $P = 0.0018$) and an increase in the fraction of wt alleles ($P = 0.0177$ and $P = 0.0108$) following treatment with 150 nM PDDX-004 and 500 nM PDDX-001 (Figs. 2A and EV2F). The same phenomenon was observed when we depleted *Exo1* genetically in the polyclonal PARG ko KB2P cell line (KB2P-P). CRISPR/Cas9-mediated targeting of *Exo1* initially resulted in a mixed population of mutated and wt *Exo1* alleles. Following 7 days of cell proliferation, there was a significant increase in the fraction of wt alleles ($P = 0.0213$) (Figs. 2B and EV2G), confirming the decreased fitness of PARG-deficient cells upon *Exo1* loss. To confirm these results, we sought to deplete *Exo1* by siRNA. Consistently, this approach resulted in a significant growth reduction in *Parg⁻/⁻* KB2P cells (KB2P-P2) (Fig. 2C). In contrast, no effect of *Exo1* targeting was observed when we transfected si*Exo1* in the newly generated BRCA2-proficient *p53⁻/⁻* mammary tumor cells with a homozygous *Parg* frameshift mutation (KP-P1) (Figs. 2C and EV2H–J). These data confirm the synthetic lethal interaction between *Exo1* and *Parg* and suggest that this phenomenon is dependent on defective BRCA2 function.

### PARG;BRCA2-deficient cells are characterized by replication defects and increased replication stress upon EXO1 depletion

To gain an insight into the mechanistic basis of EXO1 and PARG synthetic lethality in BRCA2-deficient cells, we evaluated the

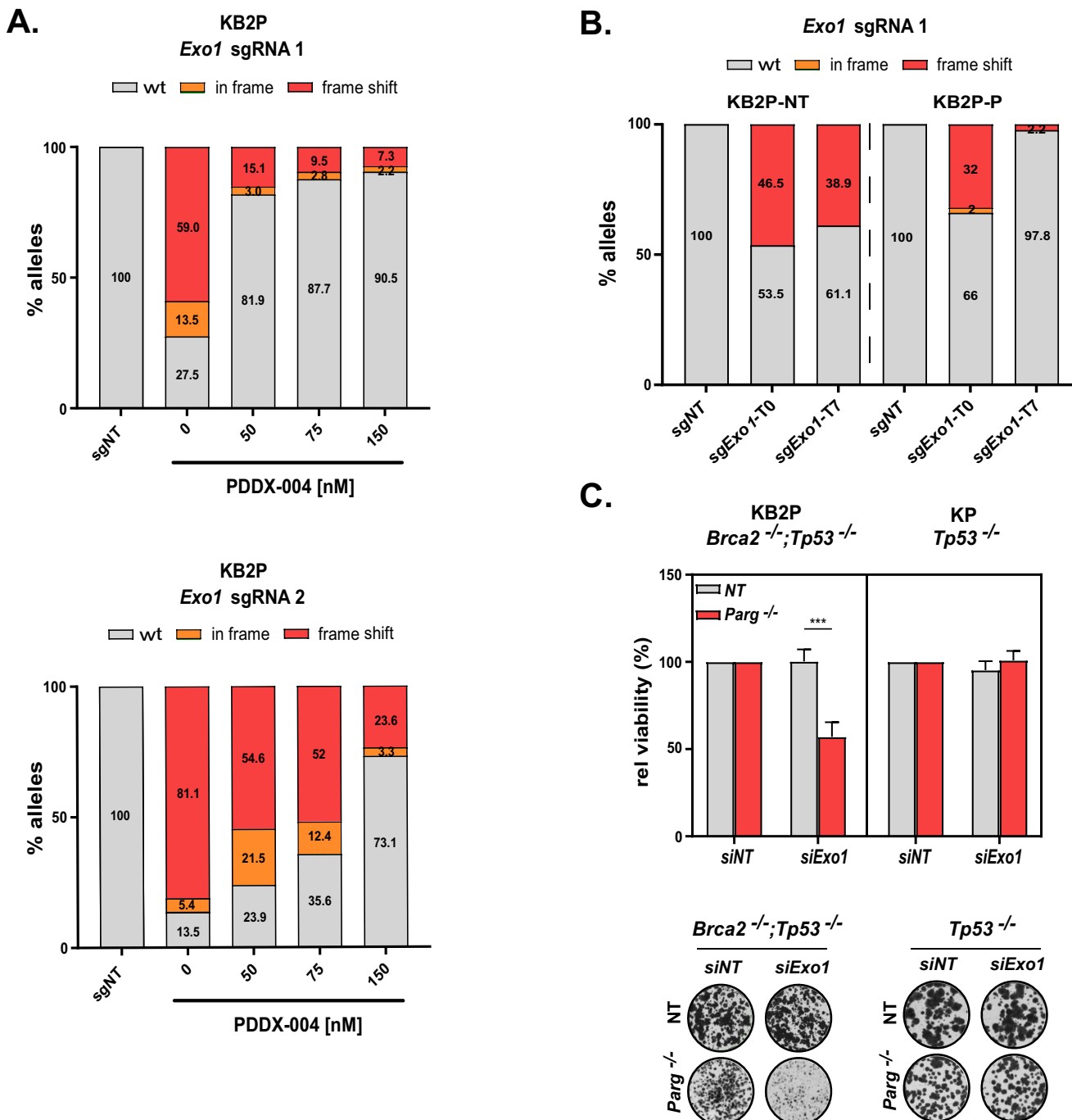

**Figure 2. PARG-deficient KB2P but not KP cells depend on EXO1.**

(A) Allelic modification rates of *Exo1* in KB2P cells upon targeting with two independent sgRNA sequences and following treatment with 0, 50, 75, and 100 nM PDDX-004 for 1 week. Data representative of median of two independent experiments, analyzed by TIDE (Brinkman et al, 2014). (B) Allelic modification rates of *Exo1* in KB2P-NT and KB2P-P2 cells upon sgRNA-mediated targeting, at day 1 after puromycin selection (T0) or after 7 days in culture (T7). Data representative of median of two independent experiments, analyzed by TIDE (Brinkman et al, 2014). (C) CTB-based viability quantification and representative images of the short-term clonogenic assay of the KB2P-NT, KB2P-P2, KP, and KP-P1 cells after targeting with *NT* or *Exo1* siRNA pools. The data are representative for three independent experiments, shown as mean ± SD of replicates, two-tailed *t* test, ***P < 0.001. Source data are available online for this figure.

effects of EXO1 and PARG co-depletion on genomic stability. For this purpose, we first measured the levels of γH2AX foci formation in KB2P-NT and KB2P-P2 cells in the presence or absence of EXO1. KB2P-P2 cells displayed a higher number of γH2AX-positive cells compared to KB2P-NT cells (Fig. 3A), whereas the siRNA-mediated *Exo1* depletion increased the number of γH2AX-positive KB2P-P2 cells up to 37% (Fig. 3A). In addition, some of the KB2P-P2 cells exhibited a pan-nuclear γH2AX signal (Fig. 3A),

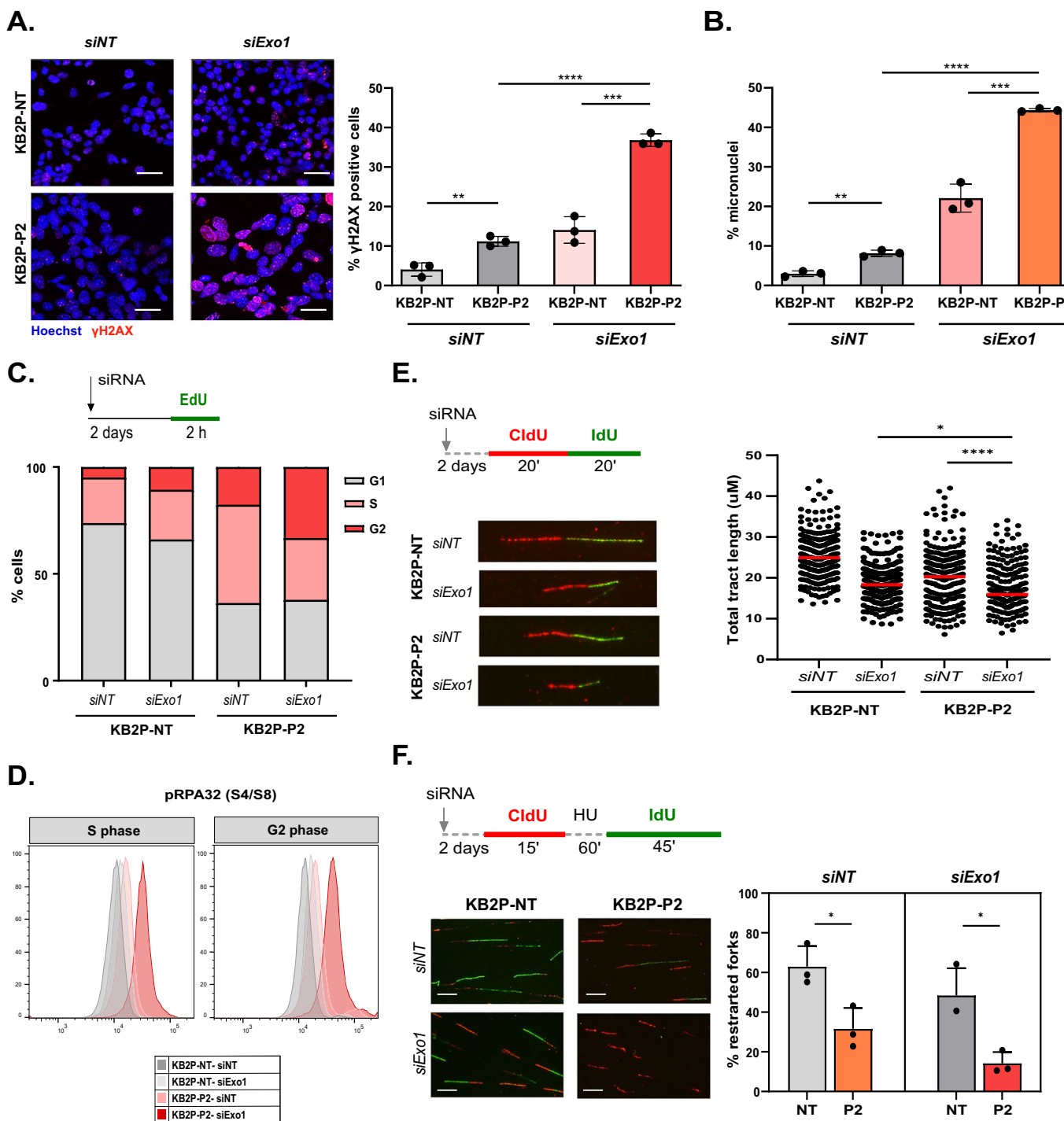

which is an indicator of lethal replication stress (Ding et al, 2016). A well-known consequence of increased replication stress is the formation of micronuclei (Xu et al, 2011; Utani et al, 2010). Indeed, the depletion of *Exo1* further augmented the already increased number of micronuclei-positive cells in the KB2P-P2 cells up to 44% (Fig. 3B). Furthermore, cell cycle analysis showed that PARG-deficient KB2P-P2 cells have a two-fold higher accumulation in S/G2 phase compared to the PARG-proficient KB2P-NT cells (Fig. 3C). Additional depletion of *Exo1* in KB2P-P2

led to the accumulation of cells in G2 phase (1.8-fold increase) (Fig. 3C). The PARG- and EXO1-deficient cells in the S/G2 phase of the cell cycle were also found to contain increased replication stress-associated ssDNA, as shown by the increased levels of the phosphorylated Replication Protein A 32 (pRPA32) at residues S4 and S8 (Fig. 3D). These observations suggest that KB2P-P2 cells carry increased replicative DNA damage but still manage to go through S/G2 phase, whereas upon *Exo1* depletion they arrest in G2.

◄ **Figure 3.   PARG;BRCA2-deficient cells are characterized by replication defects and increased replication stress upon EXO1 depletion.**

(A) IF analysis of γH2AX foci and representative images of KB2P-NT and KB2P-P2 cells, 3 days after transfection with NT and *Exo1* siRNA pools. The data are representative for three independent experiments, shown as mean ± SD of replicates, two-tailed *t* test, **$P < 0.01$, ***$P < 0.001$, ****$P < 0.0001$. Positive cells >= 10 foci. Scale bar 100 µm. (B) Percentage of micronuclei-positive KB2P-NT and KB2P-P2 cells, 3 days after transfection with NT and *Exo1* siRNA pools. The data are representative for three independent experiments, shown as mean ± SD of replicates, two-tailed *t* test, **$P < 0.01$, ***$P < 0.001$, ****$P < 0.0001$. (C) Percentage of KB2P-NT and KB2P-P2 cells 2 days after transfection with NT and *Exo1* siRNA pools in the the G1, S, and G2 phase of the cell cycle based on their co-staining for EdU and DAPI, data representative of two independent experiments, shown as mean ± SEM of $n = 2$. (D) Representative histograms of pRPA32 (S4/S3)-positive cell populations in the S and G2 phases of (C). (E) Treatment scheme, total track length analysis and representative fiber images of KB2P-NT and KB2P-P2 cells 2 days after transfection with *NT* and *Exo1* siRNA pools. The data are shown as median of three independent experiments, one-way ANOVA, *$P < 0.05$, ****$P < 0.000$. (F) Treatment scheme, analysis of the percentage of restarted forks and representative examples of stalled and restarted forks in KB2P-NT and KB2P-P2 cells 2 days after transfection with NT and *Exo1* siRNA pools. The data are representative for three independent experiments, shown as mean ± SD of replicates, Welch's *t* test, *$P < 0.05$. Scale bar 100 µm. Source data are available online for this figure.

Based on these results, we then zoomed into the effects of *Parg* and *Exo1* depletion on replication fork (RF) progression. Following the sequential labeling of cells with CldU (red) and IdU (green) for 20 min, we measured single-molecule DNA fiber tract lengths using standard protocols (Quinet et al, 2017). This showed that the depletion of either *Parg* or *Exo1* alone resulted in lower fork elongation rates in KB2P cells (Fig. 3E), indicating that PARG and EXO1 loss restrains the normal replication fork progression of BRCA2-deficient cells. However, this replication defect can still be tolerated, as most cell lines are not dependent on either of these genes (www.depmap.org), and also our *Parg*$^{-/-}$ KB2P cells form a comparable number of colonies as KB2P cells (Fig. EV1B). When both PARG and EXO1 function was impaired, cells exhibited an even more compromised RF progression (Fig. 3E), in line with the additional increase in genomic instability under these conditions (Fig. 3A–C).

PARG has been described to be important for replication fork restart (Ray Chaudhuri et al, 2015). Therefore, we then questioned whether the reduced fork elongation, observed in *Parg* and *Exo1*-depleted cells, was associated to a defect in the restart of replication forks when they stall upon DNA damage. To test this, we assessed the capability of *Parg* and/or *Exo1*-depleted KB2P cells to restart their forks following hydroxyurea (HU) treatment. Consistent with the study of Ray Chaudhuri et al (2015), *Parg* knockout (ko) cells showed a compromised ability to restart their replication forks (Fig. 3F). When *Exo1* was depleted in addition, this effect was exacerbated and the number of restarted forks dropped even further (Fig. 3F). From these data, we infer that PARG-deficient KB2P cells carry unresolved DNA lesions which restrain their replication fork progression. This phenomenon persists upon EXO1 depletion, which ultimately leads to replication fork catastrophe.

## PARG;BRCA2-deficient carry unresolved DNA lesions as the source of vulnerability for EXO1/FEN1 inhibition

Another important process regulated by PARG is the repair of SSBs (Fisher et al, 2007). Upon treatment with MMS, PARG-deficient KB2P cells are characterized by increased PAR levels (Fig. EV1C), indicative of their impaired PAR metabolism at these lesions. To elucidate the source of the persistent DNA damage that leads to replication defects in PARG;BRCA2-deficient cells, we first assessed whether the dysregulated PARylation of KB2P-P2 cells interferes with their ability to resolve PAR-regulated lesions like SSBs. Indeed, KB2P-P2 cells exhibited a compromised ability to repair SSBs when compared to the PARG-proficient KB2P-NT cells, as indicated by

their substantial increase in tail moments following MMS treatment, in alkaline comet assays (Fig. 4A). In addition, KP-P1 cells also exhibited a significant increase in their tail moments compared to KP cells (Fig. 4A). However the effect was not as prominent as in the BRCA2-deficient KB2P-P2 cells. These data support that PARG-deficient cells carry a compromised ability to repair PAR-regulated lesions, resulting in the accumulation of breaks in these cells.

We next set out to investigate which cellular process serves as the source for unresolved PAR-regulated lesions and can confer vulnerability to the loss of EXO1, in the absence of exogenous damage. In addition to *Exo1*, our genome-wide screen in *Parg*$^{-/-}$ KB2P cells identified *Fen1* as another top hit with a link to DNA repair and replication (Fig. 1B). Both *Exo1* and *Fen1* encode for 5′ flap endonucleases and 5′ to 3′ exonucleases, which is important during replication, SSBR, and Okazaki fragment processing (Keijzers et al, 2015). Therefore, we hypothesized that the vulnerability of *Parg*$^{-/-}$ KB2P cells may reside in the loss of such a nuclease activity. To elucidate this, we utilized the EXO1/FEN1 inhibitor LNT1, previously demonstrated by (Exell et al, 2016) to potently inhibit both FEN1 and EXO1 with equal potency, while showing no inhibitory effect on other proteins with conserved active sites. First, we checked whether the inhibitor-induced replication stress in a similar fashion as we observed for the siRNA-mediated *Exo1* depletion. Indeed FEN1/EXO1 inhibition resulted in increased γH2AX foci formation (Fig. EV3A), specifically in cells that are BRCA2- and PARG-deficient. To investigate the possibility that this is an effect of PARP1 trapping, similar to the toxic effect observed for PARP inhibitors, we measured the levels of chromatin-bound PARP1 in KB2P-NT and KB2P-P2 cells upon LNT1 treatment, using a trapping assay that was previously described (Murai et al, 2012). The immunoblot analysis showed that although olaparib treatment is indeed inducing increased association of PARP1 to the chromatin, LNT1 treatment was not giving the same effect and PARP1 chromatin-associated levels were rather similar to the untreated KB2P-P2 cells (Fig. EV3B). Since PARG-deficient cells are characterized by increased PARylation, we next investigated whether EXO1/FEN1 inhibition had any effect on the PAR levels in *Parg*$^{-/-}$ cells. Indeed, LNT1 inhibitor treatment induced a significant increase in the PAR levels in *Parg*$^{-/-}$ cells, both BRCA2-proficient and -deficient (Figs. 4C and EV3C). PAR levels were also significantly higher in KB2P-P2 cells compared to KP-P1 cells (Figs. 4B and EV3C), consistent with the different levels of DNA damage in the two cell lines following EXO1/FEN1 inhibition (Fig. EV3A). In agreement with this, Western blot analysis for pCHK1 (S317), pRPA2 (S4/8),

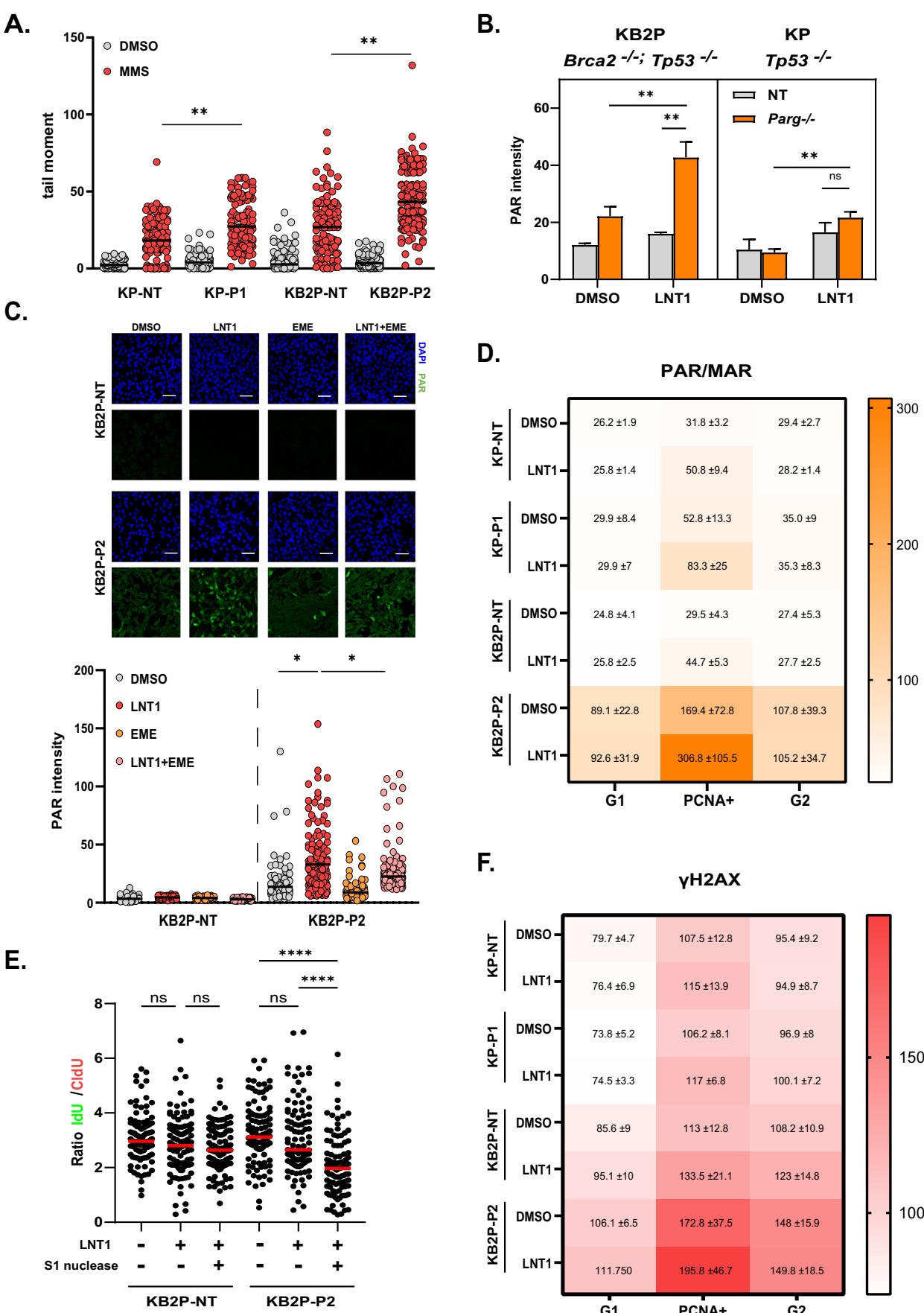

**Figure 4. PAR-regulated ssDNA lesions as the source of the vulnerability to EXO1/FEN1 inhibition.**

(A) ssDNA breaks in genomic DNA quantified by alkaline comet assays in KP-NT, KP-P1, KB2P-NT, and KB2P-P2 cells following 15 min incubation with 0.01% MMS. Comet tail moments were scored following staining of genomic DNA with SYBR Green. For each sample, scatter plots are the tail moments of about 300 individual cells combined from $n = 3$ experiments. Data shown as median of three independent experiments, two-tailed $t$ test of means **$P < 0.01$. (B) Intensity quantification of PAR immunofluorescence in KB2P-NT, KB2P-P2, KP-NT and KP-P1 cells after 2 h treatment with 10 µM LNT1. Data are representative of three independent experiments shown as mean ± SD of $n = 3$, two-tailed $t$ test, ns non-significant, **$P < 0.01$. (C) Intensity quantification and representative images of PAR immunofluorescence in KB2P-NT and KB2P-P2 cells 2 h after treatment with 10 µM LNT1 and/or 2 µM emetine. Data are representative of three independent experiments, shown as median of $n = 3$, two-tailed $t$ test on means, ns non-significant, *$P < 0.05$. Scale bar 50 µm. (D) Heatmap representing the ScanR quantification of median nuclear intensities of anti-ADP-ribose (PAR/MAR)) immunofluorescence of KB2P-NT, KB2P-P2, KP-NT and KP-P1 cells following 30 min incubation with 10 µM LNT1. Data are shown as mean ± SD of $n = 4$. (E) Analysis of the ratio of IdU/CldU tract lengths of replication forks in KB2P-NT and KB2P-P2 cells following combinational treatment with 10 µM LNT1 or sequential treatment with S1 nuclease buffer, as indicated. The data are shown as median of two independent experiments, one-way ANOVA, ns non-significant, ****$P < 0.000$. (F) Heatmap representing the ScanR quantification of median nuclear intensities of γH2AX immunofluorescence of KB2P-NT, KB2P-P2, KP-NT, and KP-P1 cells following 30 min incubation with or without 10 µM LNT1. Data are shown as mean ± SD of $n = 3$. Source data are available online for this figure.

and γH2AX showed increased expression in the samples with elevated ADP-ribose (MAR/PAR) levels (Fig. EV3D). These results indicate that the DNA lesions, which lead to elevated replication stress, specifically in BRCA2/PARG-deficient cells following EXO1/FEN1 inhibition, require a PAR-mediated process to be resolved.

One of the main sites for PAR metabolism in unperturbed cells are unligated Okazaki fragments (Hanzlikova et al, 2018). In this context, FEN1 inhibition can increase PARylation levels because of the accumulation of unligated Okazaki fragments (Hanzlikova et al, 2018). Therefore, we evaluated whether the LNT1-induced increase in PARylation could be reversed by Okazaki fragment formation suppression using emetine. Consistent with this hypothesis, emetine treatment was able to partially reverse the increased PARylation levels in the LNT1-treated $Parg^{-/-}$ KB2P cells (Fig. 4C), supporting that at least part of the unresolved lesions accumulating in these cells is a result of unligated Okazaki fragments.

Since Okazaki fragment processing takes place during replication, we then investigated whether these PAR-decorated lesions are specifically enriched in the S phase of the cell cycle. To quantify different cell cycle phases, we applied high-content indirect immunofluorescence using PCNA staining to determine G1 (2n, PCNA-), S phase (PCNA + ), and G2 (4n, PCNA-) cells. Nuclear ScanR analysis indeed confirmed that ADP-ribosylation (PAR/MAR) levels are higher in KB2P-P2 cells, compared to KB2P-NT cells (two-way ANOVA; $P = 0.0204$), KP-NT ($P = 0.0281$) or KP-P1 cells ($P = 0.0196$)and they remain the highest during S phase (Fig. 4D). This indicates that KB2P-P2 cells carry persistent PAR-decorated lesions along the different cell cycle phases and unligated Okazaki fragments may consist the main source. Treatment with LNT1 resulted in a pronounced increase in ADP-ribose levels during the S phase in KB2P-P2 cells (Student's $t$ test; $P = 0.0253$) (Fig. 4D), suggesting that the observed surge in PARylation upon EXO1/FEN1 inhibition may be attributed to an accumulation of unligated Okazaki fragments in these cells. In agreement, the PAR-mediated recruitment of the scaffold protein XRCC1, which is an important step in the PAR-mediated Okazaki fragment processing during S phase, increased upon EXO1/FEN1 inhibition of PARG-deficient cells KP and KB2P cells (Fig. EV3E), (Student's $t$ test; $P = 0.0019$ for KP-P1 and $P = 0.0213$ for KB2P-P2). To corroborate that the heightened SSB signaling occurring upon EXO1/FEN1 inhibition originates from ssDNA gap accumulation during DNA replication, we performed a DNA fiber assay with S1 nuclease treatment, as detailed by Quinet et al (2017). Notably, ssDNA-specific cleavage of the replication forks with S1 nuclease led to a significant reduction of the fiber length specifically in LNT1-treated KB2P-P2 cells (Fig. 4E). This finding confirms that ssDNA gaps are the predominant source of damage in these cells. Further, to determine whether the increase of ssDNA gaps during replication correlates with a rise in DSBs, we performed high-content indirect immunofluorescence analysis to measure γH2AX levels across different cell cycle phases. Consistent with our earlier observations, LNT1 treatment resulted in a significant increase in γH2AX levels specifically during S phase, in KB2-P2 cells (Student's $t$ test; $P = 0.0164$), paralleling the pattern observed in their ADP-ribosylation levels under these conditions (Fig. 4F). These results suggest that PARG;BRCA2-deficient cells achieve a functional PAR signaling cascade during S phase, but they are unable to resolve the unligated Okazaki fragments occurring upon EXO1/FEN1 inhibition, leading to ssDNA gap accumulation during replication and ultimately to the formation of DSBs.

## PARG;BRCA2-deficient cells require both FEN1 and EXO1 activity for their survival

The resolution of ssDNA lesions is critical for the survival of HR-deficient cells since the conversion of SSBs or gaps into DSBs during DNA replication causes toxicity due to inappropriate DSB repair (Kuzminov, 2001; Cortés-Ledesma and Aguilera, 2006). Therefore, we hypothesized that EXO1/FEN1 inhibition using LNT1, inhibits the growth of PARG;BRCA2-deficient cells. Indeed, LNT1 treatment strongly affected KB2P-P2 cell viability, confirming the dependence of these cells on EXO1 and FEN1 function (Fig. 5A). In addition, in this assay the sensitivity depended mainly on the BRCA2 deficiency, since the LNT1 inhibitor treatment of the BRCA2-proficient KP-P1 cells did not show a clear effect (Fig. 5A). To exclude the possibility that the viability effect was only a result of EXO1 inhibition, we then independently performed an siRNA-mediated knockdown of Fen1. Similar to Exo1 depletion, siRNA-mediated knockdown of Fen1 resulted in decreased cellular viability specifically in $Parg^{-/-}$ BRCA2;p53-deficient cells, whereas no effect on cell viability was observed in $Parg^{-/-}$ p53-deficient cells (Fig. 5B). To test whether FEN1 and EXO1 are epistatic, we also compared the single siRNA-mediated knockdown with the knockdown of both Fen1 and Exo1. As a result, we found a significant decrease in the survival of $Parg^{-/-}$ KB2P cells when both genes are depleted (Fig. 5C). This result may be explained by a redundancy of these two enzymes in DNA repair mechanisms. Importantly, the Fen1 and Exo1 depletion effect on cell viability was only present in the $Parg^{-/-}$ KB2P cells (Fig. 5C).

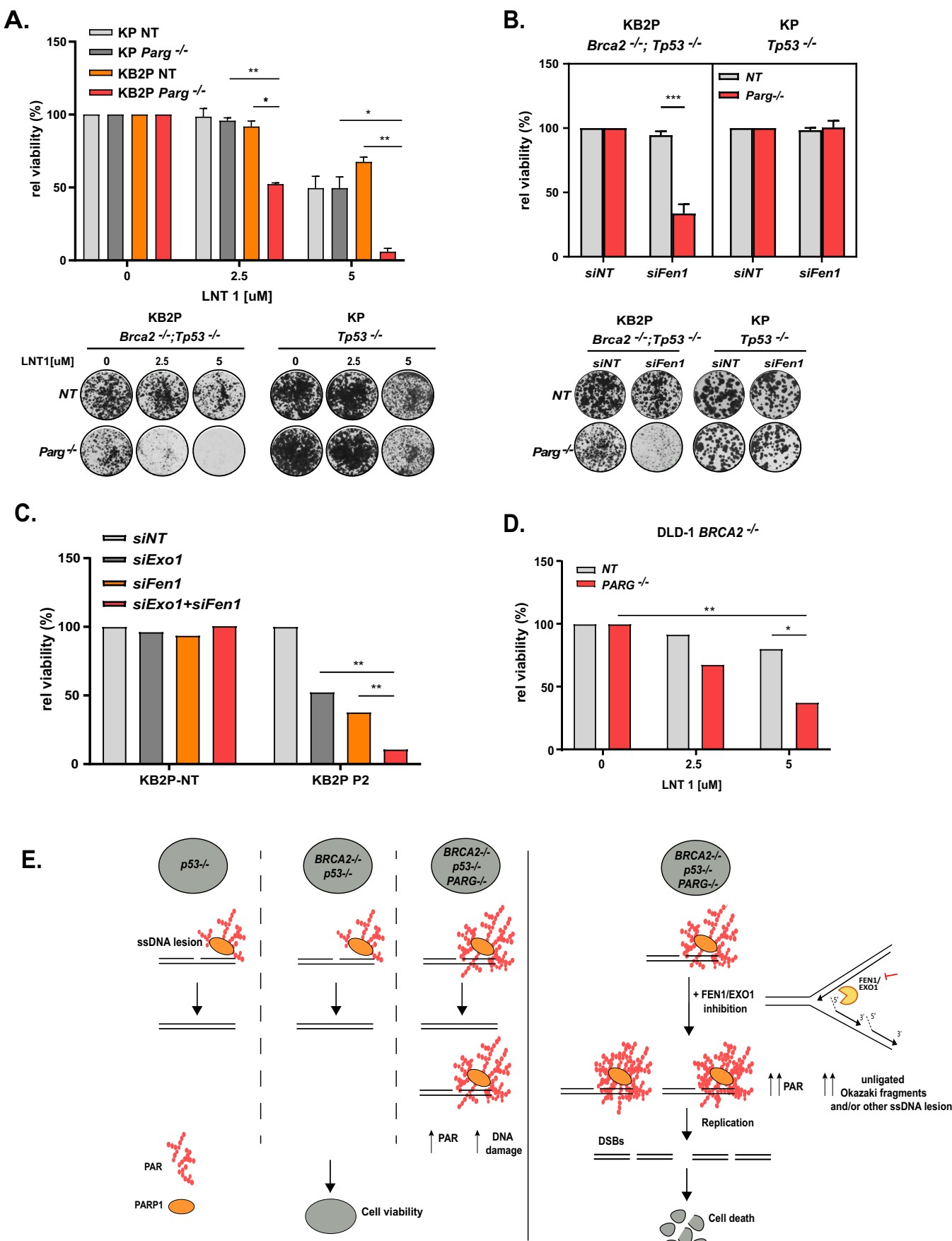

◀ **Figure 5.  Both EXO1 and FEN1 depletion are synthetic lethal specifically for PARG;BRCA2;p53-deficient cells.**

(**A**) CTB-based viability quantification and representative images of the long-term clonogenic assay of KB2P-NT, KB2P-P2, KP-NT, and KP-P1 cells in the presence of 0, 2.5, and 5 μM of LNT1. The data are representative for three independent experiments, shown as mean ± SD of replicates, two-tailed t test, ns, non-significant, *P < 0.05, **P < 0.01. (**B**) CTB-based viability quantification and representative images of the short-term clonogenic assay of the KB2P-NT, KB2P-P2 (Parg$^{-/-}$), KP-NT, and KP-P1 (Parg$^{-/-}$) cells after targeting with NT or Fen1 siRNA pools. The data are representative for three independent experiments, shown as mean ± SD of replicates, two-tailed t test, ***P < 0.001. (**C**) CTB-based viability quantification of the short-term clonogenic assay of the KB2P-NT and KB2P-P2 cells after targeting with NT, Exo1, Fen1, Fen1+Exo1 siRNA pools, data representative of averages of three technical replicates of two independent experiments, shown as mean ± SD of n = 2, two-tailed t test, **P < 0.01. (**D**) CTB-based viability quantification of the long-term clonogenic assay of the DLD-1 BRCA2 (−/−) and DLD-1 BRCA2 (−/−) PARG ko cells in the presence of 0, 2.5, and 5 μM of LNT1. Data are representative of averages of three technical replicates of two independent experiments, shown as mean ± SD of n = 2, two-tailed t test, *P < 0.05, **P < 0.01. (**E**) Proposed model illustrating the survival dependency of PARG;BRCA2;p53-deficient cells on EXO1 and FEN1 function. Source data are available online for this figure.

In addition to BRCA2, BRCA1 is a well-known player in the HR pathway (Chen et al, 2018). To test whether the FEN1/EXO1 inhibition affects BRCA1-deficient cells to a similar extent, we established PARG-proficient (KB1P-NT) and -deficient KB1P (KB1P-P) cells (Fig. EV4A,B), treated them with the LNT1 inhibitor and assessed their viability. Although the PARG-deficient cells were more sensitive at higher concentrations, also the PARG-proficient KB1P cells were sensitive to increasing LNT1 concentrations (Fig. EV4C). This result is consistent with the recent findings of (van de Kooij et al, 2024), suggesting that there is a synthetic lethal interaction between BRCA1 and EXO1.

Importantly, in previous studies, we found that PARG deficiency causes PARPi resistance in human BRCA2-deficient tumor cells and may contribute to PARPi resistance in patients (Gogola et al, 2018). Therefore, we tested whether the LNT1 inhibitor also kills human cancer cells carrying a PARG deficiency. For this purpose, we generated polyclonal PARG- and BRCA2-deficient DLD-1 cells. Successful PARG depletion was validated by TIDE (Tracking of Indels by Decomposition) analysis (Fig. EV4D) and functionally by reduced sensitivity to-/olaparib and increased MMS-induced PARylation levels in PARG;BRCA2-deficient DLD-1 cells compared to BRCA2-deficient DLD-1 cells (Fig. EV4E,F). When we treated BRCA2-deficient DLD-1 cells with the LNT1 inhibitor, we observed a significant decrease in cellular viability upon PARG loss (Fig. 5D). Hence, the increased dependence of PARG;BRCA2-deficient cells on EXO1/FEN1 function is also relevant in human tumor cells.

These observations are consistent with the model that EXO1/FEN1 inhibition elevates ADP-ribosylation levels in PARG;BRCA2-deficient cells, likely due to persisted PARP1 auto-PARylation on unligated Okazaki fragments. These fragments, remaining as ssDNA gaps, have the potential to evolve into DSBs upon replication. Such DSBs are tolerated in p53-deficient cells, but become lethal in the context of compromised homology-directed DNA repair due to the concurrent loss of BRCA2 (Fig. 5E). Consequently, we deduce that the survival of PARG;BRCA2-deficient cells is critically contingent upon the functional integrity of EXO1 and FEN1 nucleases.

## Discussion

In this study, we found that PARG;BRCA2;p53-deficient cells depend on EXO1 and FEN1. In particular, EXO1 came out as a major hit in two conceptually different, genome-wide CRISPR/

Cas9 screens, in which we either depleted Parg genetically or we inhibited PARG chemically. Our data underpin the power of genome-wide CRISPR/Cas9 screens to dissect the molecular dependencies associated with the deficiency of a key player in posttranslational modifications, in this case PAR biology. In addition to EXO1 and FEN1, we have charted the landscape of genes that become essential when PARG is depleted. Besides DNA metabolism-associated processes that include DNA replication, MMR, NER, and BER, other pathways affecting RNA, ribosome, and spliceosome biology were identified. We believe that this knowledge is important for at least two reasons: (1) PARGi are currently developed for the treatment of cancers with frequent HR deficiencies like ovarian cancer (Chen and Yu, 2019; Pillay et al, 2021; Slade, 2020), and we have a poor understanding which genetic deficiencies of cancer cells can be exploited with these new compounds. In the era of precision medicine, this may provide useful information to guide future clinical trials. (2) Based on our unique genetically engineered mouse models to study PARPi resistance in BRCA1/2-mutated tumors, we expect that PARG loss may be a mechanism of secondary resistance that is selected out when patients receive continuous, daily PARPi treatment, e.g., as maintenance therapy (Gogola et al, 2018; Bhin et al, 2023). Although the relevance of this mechanism still needs to be validated using second biopsies of patient tumors that initially responded and eventually became drug resistant, our data support the use of EXO1/FEN1 inhibitors to counteract resistance mediated by PARG loss. Once EXO1/FEN1 inhibitors have been developed that achieve sufficient pharmacokinetic and dynamic properties in vivo, they may also be useful when combined with PARPi and thereby prevent or delay drug resistance.

To understand the mechanistic basis of the EXO1/FEN1 dependency of PARG;BRCA2;p53-deficient cells, we focused here on the study of their DNA replication, SSBR, and Okazaki fragment processing vulnerabilities, which constitute important PAR-regulated processes and in which EXO1 and FEN1 are involved.

Our experiments show that Parg$^{-/-}$;Brca2$^{-/-}$ tumor cells are characterized by increased DNA damage, replication stress and low potency to restart their replication forks, in line with what was described by Chaudhuri et al, 2015. Upon replication stress, replication forks stall and reverse until the impediment is fixed, which will then allow them to restart (Lemaçon et al, 2017). Since fork restart is inhibited by PARylation (Lemaçon et al, 2017), PARG loss can slow down their restart and the replication process (Pillay et al, 2021). However, this does not seem to be lethal for the cells unless EXO1 is also depleted. The replication stress profile of Parg$^{-/-}$;Brca2$^{-/-}$ cells is exacerbated upon Exo1 siRNA-targeting, as

a result of the accumulation of additional DNA damage that restricts their ability to progress through the cell cycle.

In our effort to elucidate the source of DNA damage in *Parg⁻/⁻;Brca2⁻/⁻* cells, we found that they carry a compromised ability to repair SSBs upon exogenous DNA damage induction. This finding provides evidence that the impaired de-PAR-ylation mechanism of PARG-deficient cells restrains their ability to resolve PAR-regulated lesions. One of the initial steps during SSBR is the PAR-mediated recruitment of XRCC1 to the nicked DNA. XRCC1 forms a complex with LIG3 and other repair factors to carry out the filling and ligation of the DNA. A previous study suggested that PAR removal is important to allow the XRCC1 complex to perform this process (Wei et al, 2013). It is possible that excessive PARylation may prevent the repair complex to reach the break site or may prevent the DNA repair protein release from the DNA and thus their availability for recruitment to other lesions. Such a scenario may explain why SSBs remain unresolved in the PARG-deficient background.

A similar PAR-regulated pathway takes place during the processing of unligated Okazaki fragments. Auto-PARylation of PARP1 serves as a signal for the recruitment of the SSBR protein XRCC1 in an alternative Okazaki processing pathway, when FEN1 function is abrogated (Hanzlikova et al, 2018; Vaitsiankova et al, 2022). FEN1 is the 5' flap exonuclease involved in the classical Okazaki fragment maturation and since EXO1 also carries a similar 5' flap endonuclease activity, it may also contribute to flap removal during this process (Keijzers et al, 2015). Here, we observed EXO1/FEN1 inhibition with the dual inhibitor LNT1 (Ward et al, 2017), results in elevated S-phase ADP-ribose levels in PARG;BRCA2-deficient cells, which are mainly associated to unligated Okazaki fragments. The unligated Okazaki fragments show to persist in PARG-deficient cells as ssDNA gaps upon EXO1/FEN1 inhibition.

Persistent ssDNA breaks or ssDNA gaps become lethal to BRCA2-deficient cells when they are converted into DSBs during replication, because the lesions cannot be accurately resolved in the absence of HR (Kuzminov, 2001; Cortés-Ledesma and Aguilera, 2006). Indeed, we show that LNT1 inhibition sensitizes cells to PARG loss, specifically when they are BRCA2-deficient. In addition, our results demonstrate that both *Exo1* and *Fen1* are individually essential for the survival of PARG;BRCA2-deficient cells. Since we do not see complete epistasis of *Exo1* and *Fen1* knockdown on cell survival, we cannot exclude that EXO1 and FEN1 protect *Parg⁻/⁻* cells by independent, and yet-to-be-determined mechanisms. In general, it would be interesting to define the distinct roles of EXO1 versus FEN1 and test which of these contribute to the increased sensitivity of *Parg⁻/⁻* cells. It is also possible that there is redundancy in their function, but the full

activity of both enzymes is required to deal with the lesions induced by the PARG deficiency.

Van de Kooij et al (2024) recently reported that even in the presence of functional PARG, EXO1 is essential for BRCA1-deficient cells. Their data suggest that the underlying defect is the impaired single-strand annealing, a DSB repair pathway that involves both BRCA1 and EXO1 (van de Kooij et al, 2024). Interestingly, they did not report a similar dependence of BRCA2-deficient cells on EXO1, which is consistent with our findings (Bhin et al, 2023) that demonstrate distinct capacities of BRCA1- and BRCA2-deficient cells in restoring HR. Consistent with their data, our study also reveals an increased sensitivity of PARG-proficient BRCA1;p53-deficient cells to LNT1. Furthermore, Mengwasser et al (2019) described a synthetic lethal interaction between FEN1 and BRCA2 deficiency. Although our experiments did not show marked sensitization of BRCA2-deficient cells to FEN1 depletion in the presence of active PARG, it is plausible that HR-deficient cells generally exhibit a strong reliance on EXO1 and FEN1-mediated DSB repair pathways, particularly when their ssDNA repair mechanisms are compromised.

Altogether, our data provide the rationale for the development of clinically relevant EXO1/FEN1 inhibitors. Inhibiting these nucleolytic enzymes may offer a promising tool to target HR-deficient tumors as well as PARPi resistance caused by PARG deficiency. In addition, major efforts are put in the design of PARG inhibitors to treat tumors with replication defects (Pillay et al, 2019), and we expect that the addition of EXO1/FEN1 inhibition may further increase their efficacy. Notably, PARG inhibitor compounds are already advancing into Phase 1 clinical trials for HR-deficient solid tumors (https://prn.to/43Jn6zU). Our study shows that PARG inhibition using PDDX inhibitors induces toxicity in PARG;BRCA2-deficient cells through heightened ADP-ribosylation levels. This observation, potentially attributable to off-target effects, highlights the critical need for meticulous compound evaluation to ensure the effective and safe implementation of PARG inhibitors in the clinic. Together with our genetic map of essential genes in PARG-deficient cells, we believe that our data will contribute to the optimization and efficient application of PARG inhibitors in precision oncology.

## Methods

### Reagents and tools

See Table 1.

**Table 1. Reagents and tools.**

| Reagent/resource | Reference or source | Identifier or catalog number |
|---|---|---|
| **Experimental models** | | |
| *List cell lines, model organism strains, patient samples, isolated cell types etc. Indicate the species when appropriate.* | | |
| KB2P1.21 | Evers et al, 2008 | N/A |
| KP3.33 | Evers et al, 2008 | N/A |
| KB1P | Evers et al, 2008 | N/A |
| HEK293FT | ATCC | RRID:CVCL_6911 |
| DLD BRCA2−/− | Horizon Discovery | #HD 105-007; RRID:CVCL_HD57 |

**Table 1.** (continued)

| Reagent/resource | Reference or source | Identifier or catalog number |
|---|---|---|
| **Recombinant DNA** | | |
| *Indicate species for genes and proteins when appropriate* | | |
| Mouse CRISPR Knockout Pooled Library YUSA v2 (lentiviral) | Tzelepis et al, 2016 | N/A |
| piKRUNC-Puro-PARG_sgRNA1 (lentiviral) | This paper | N/A |
| piKRUNC-Puro- (non-targeting) NT_sgRNA (lentiviral) | This paper | N/A |
| plentiGuide v2_EXO1_sgRNA1 (lentiviral) | This paper | N/A |
| plentiGuide v2_EXO1_sgRNA2 (lentiviral) | This paper | N/A |
| lentiGuide-Puro (non-targeting) NT sgRNA (lentiviral) | This paper | N/A |
| **Antibodies** | | |
| Mouse monoclonal anti-PAR (10H) | Millipore | Cat# ab1791 |
| Rabbit anti-MAR/PAR | Cell Signaling | Cat# 83732 |
| Rabbit polyclonal anti-PARP1 | Cell Signaling | Cat# 9542 |
| Rabbit polyclonal anti-Phospho Histone H2A.X (Ser139) | Cell Signaling | Cat# 2577 |
| Mouse anti-Phospho Histone H2A.X (Ser139) | Millipore | Cat# 05-636 |
| Rabbit monoclonal anti-phospho RPA32 (S4/S8) | Bethyl Laboratories | Cat# A700-009 |
| Rabbit anti-PCNA | Abcam | Cat# ab252848 |
| Mouse anti-PCNA | Santa Cruz | Cat# sc-56 |
| Rat monoclonal anti-BrdU/CldU | Abcam | Cat# ab6326 |
| Mouse monoclonal anti-BrdU/IdU | Becton Dickinson | Cat# 347580 |
| Goat polyclonal anti-mouse, Alexa Fluor 488-conjugated | Invitrogen | Cat# A11001 |
| Goat polyclonal anti-rabbit, Alexa Fluor 568-conjugated | Invitrogen | Cat# A11011 |
| Goat polyclonal anti-rabbit, HRP conjugated | DAKO | Cat# P0448 |
| Goat polyclonal anti-mouse, HRP conjugated | DAKO | Cat# P0447 |
| **Oligonucleotides and sequence-based reagents** | | |
| ON-TARGETplus Non-targeting siRNA #1 | Dharmacon/Horizon | Cat# D-001810-01 |
| ON-TARGETplus Mouse Exo1 (26909) siRNA - SMARTpool | Dharmacon/Horizon | Cat# L-060591-01 |
| ON-TARGETplus Mouse Fen1 (14156) siRNA - SMARTpool | Dharmacon/Horizon | Cat# L-064652-01 |
| NT(non-targeting) sgRNA (CRISPR/Cas9 genome editing) | Integrated DNA Technologies (IDT) | TGATTGGGGGTCGTTCGCCA |

**Table 1.** (continued)

| Reagent/resource | Reference or source | Identifier or catalog number |
|---|---|---|
| Mouse *Parg* sgRNA #1 (exon 3) (CRISPR/Cas9 genome editing) | Integrated DNA Technologies (IDT) | CAGCTTAGTAATGCTAACAT |
| Mouse *Parg* sgRNA #2 (exon 3) (CRISPR/Cas9 genome editing) | Integrated DNA Technologies (IDT) | TTTGGTAGCGGAGTCCCCGC |
| Mouse *Parg* sgRNA #2 (exon 9) (CRISPR/Cas9 genome editing) | Integrated DNA Technologies (IDT) | TTCATCTTGGCATTCCGTCG |
| Human *Parg* sgRNA (exon 7) (CRISPR/Cas9 genome editing) | Integrated DNA Technologies (IDT) | TGCTATTCTGAAATACAATG |
| Mouse Exo1 sgRNA #1 (exon 5) (CRISPR/Cas9 genome editing) | Integrated DNA Technologies (IDT) | ACCCCGATATCGTGAAGGT |
| Mouse Exo1 sgRNA #2 (exon 5) (CRISPR/Cas9 genome editing) | Integrated DNA Technologies (IDT) | ACCCCGATATCGTGAAGG |
| 5′ Biotin-TEG capture oligo NdeI (gDNA capture) | Integrated DNA Technologies (IDT) | TGCTTACCGTAACTTGAAAGTATTTCGATTTCTTGGCTTTATATATCTTG |
| 5′ Biotin-TEG capture oligo XhoI (gDNA capture) | Integrated DNA Technologies (IDT) | GATCTAGATGGATGCAGGTCGAAAGGCCCGGAGATGAGGAAGAGGAGAAC |
| Mouse *Parg* sgRNA 1 forward primer (TIDE analysis) | Integrated DNA Technologies (IDT) | TCACAGGGCAAACGTCTCAC |
| Mouse *Parg* sgRNA 1 reverse primer (TIDE analysis) | Integrated DNA Technologies (IDT) | TCCAGTTCCAATGTCCTCGG |
| Mouse *Parg* sgRNA 2 forward primer (TIDE analysis) | Integrated DNA Technologies (IDT) | GCAAACGTCTCACTATGGCG |
| Mouse *Parg* sgRNA 2 reverse primer (TIDE analysis) | Integrated DNA Technologies (IDT) | AACACGGGCACTGGAGTTAC |
| Mouse *Exo1* sgRNA 1 forward primer (TIDE analysis) | Integrated DNA Technologies (IDT) | TGCCTTTCCGAGCTTTGACCA |
| Mouse *Exo1* sgRNA 1 reverse primer (TIDE analysis) | Integrated DNA Technologies (IDT) | CCAGGCTGCCTTTAAACTCGC |
| Mouse *Exo1* sgRNA 2 forward primer (TIDE analysis) | Integrated DNA Technologies (IDT) | CACTTTCCAGTGTCAGGGGTGG |
| Mouse *Exo1* sgRNA 2 reverse primer (TIDE analysis) | Integrated DNA Technologies (IDT) | TTTCTGCTACACCAGGCACAGC |
| Mouse YUSA library barcoded forward primer 1 (Library amplification) | Integrated DNA Technologies (IDT) | ACACTCTTTCCCTACACGACGCTCTTCCGATCTCGTGATGGCTTTATATATCTTGTGGAAAGGACG |
| Mouse YUSA library barcoded forward primer 2 (Library amplification) | Integrated DNA Technologies (IDT) | ACACTCTTTCCCTACACGACGCTCTTCCGATCTACATCGGGCTTTATATATCTTGTGGAAAGGACG |
| Mouse YUSA library barcoded forward primer 3 (Library amplification) | Integrated DNA Technologies (IDT) | ACACTCTTTCCCTACACGACGCTCTTCCGATCTGCCTAAGGCTTTATATATCTTGTGGAAAGGACG |
| Mouse YUSA library barcoded forward primer 4 (Library amplification) | Integrated DNA Technologies (IDT) | ACACTCTTTCCCTACACGACGCTCTTCCGATCTTGGTCAGGCTTTATATATCTTGTGGAAAGGACG |
| Mouse YUSA library barcoded forward primer 5 (Library amplification) | Integrated DNA Technologies (IDT) | ACACTCTTTCCCTACACGACGCTCTTCCGATCTCACTGTGGCTTTATATATCTTGTGGAAAGGACG |
| Mouse YUSA library barcoded forward primer 6 (Library amplification) | Integrated DNA Technologies (IDT) | ACACTCTTTCCCTACACGACGCTCTTCCGATCTATTGGCGGCTTTATATATCTTGTGGAAAGGACG |

**Table 1.** (continued)

| Reagent/resource | Reference or source | Identifier or catalog number |
| --- | --- | --- |
| Mouse YUSA library barcoded forward primer 7 (Library amplification) | Integrated DNA Technologies (IDT) | ACACTCTTTCCCTACACGACGCTCTTCCGATCTGATCTGGGCTTTATATATCTTGTGGAAAGGACG |
| Mouse YUSA library barcoded forward primer 8 (Library amplification) | Integrated DNA Technologies (IDT) | ACACTCTTTCCCTACACGACGCTCTTCCGATCTTCAAGTGGCTTTATATATCTTGTGGAAAGGACG |
| Mouse YUSA library barcoded forward primer 9 (Library amplification) | Integrated DNA Technologies (IDT) | ACACTCTTTCCCTACACGACGCTCTTCCGATCTCTGATCGGCTTTATATATCTTGTGGAAAGGACG |
| Mouse YUSA library barcoded forward primer 10 (Library amplification) | Integrated DNA Technologies (IDT) | ACACTCTTTCCCTACACGACGCTCTTCCGATCTAAGCTAGGCTTTATATATCTTGTGGAAAGGACG |
| Mouse YUSA library barcoded forward primer 11 (Library amplification) | Integrated DNA Technologies (IDT) | ACACTCTTTCCCTACACGACGCTCTTCCGATCTGTAGCCGGCTTTATATATCTTGTGGAAAGGACG |
| Mouse YUSA library barcoded forward primer 12 (Library amplification) | Integrated DNA Technologies (IDT) | ACACTCTTTCCCTACACGACGCTCTTCCGATCTTACAAGGGCTTTATATATCTTGTGGAAAGGACG |
| Mouse YUSA library reverse primer (Library amplification) | Integrated DNA Technologies (IDT) | GTGACTGGAGTTCAGACGTGTGCTCTTCCGATCTGAAAAGCGCCTCCCCTACC |
| Mouse *Parg* forward primer (RT-qPCR) | Integrated DNA Technologies (IDT) | GGTGCCGAGGACATTGGAA |
| Mouse *Parg* reverse primer (RT-qPCR) | Integrated DNA Technologies (IDT) | TCAAATGGAGGCGAATCACCT |
| Mouse *Exo1* forward primer (RT-qPCR) | Integrated DNA Technologies (IDT) | ATGGGGATTCAAGGGTTACTTCA |
| Mouse *Exo1* reverse primer (RT-qPCR) | Integrated DNA Technologies (IDT) | AGCCAACAGTAGGTATCCACAG |
| Mouse *Fen1* forward primer (RT-qPCR) | Integrated DNA Technologies (IDT) | TTCACGGCCTTGCCAAACTAA |
| Mouse *Fen1* reverse primer (RT-qPCR) | Integrated DNA Technologies (IDT) | ACAGCAATCAGGAACTGGTAGA |
| Mouse *Hprt* forward primer (RT-qPCR) | Integrated DNA Technologies (IDT) | CTGGTGAAAAGGACCTCTCG |
| Mouse *Hprt* reverse primer (RT-qPCR) | Integrated DNA Technologies (IDT) | TGAAGTACTCATTATAGTCAAGGGCA |
| **Chemicals, enzymes, and other reagents** | | |
| AZD2281 (olaparib), PARP inhibitor | Syncom, Groningen,The Netherlands | CAS: 763113-22-0 |
| PDDX-001 (PDD00017273), PARG inhibitor | Drug Discovery Unit, Manchester University | CAS: 1945950-21-9 |
| PDDX-004 (PDD00017272), PARG inhibitor | Syncom, Groningen, The Netherlands | Batch# MLN30949-30954-J |
| Methyl methanesulfonate (MMS) | Sigma-Aldrich | Cat# 129925 |
| BD Phosflow Fix Buffer 1 | BD Biosciences | Cat# 557870 |
| BD Phosflow Perm Buffer III | BD Biosciences | Cat# 558050 |
| DharmaFECT 1 Transfection Reagent | Dharmacon/Horizon | Cat# T-2001-01 |
| EnGen Spy Cas9 NLS | New England Biolabs | Cat# M0646 |
| Lipofectamine RNAiMAX Reagent | Invitrogen | Cat# 13778 |
| XhoI restriction enzyme | New England Biolabs | Cat# R0146 |

**Table 1.** (continued)

| Reagent/resource | Reference or source | Identifier or catalog number |
|---|---|---|
| Ndel restriction enzyme | New England Biolabs | Cat# R0111 |
| BsmBI-v2 | New England Biolabs | Cat# R0739 |
| rCutSmart buffer | New England Biolabs | Cat# B6004 |
| Exonuclease I | New England Biolabs | Cat# M0293 |
| Exonuclease I Reaction Buffer | New England Biolabs | Cat# B0293 |
| Phusion High-Fidelity DNA Polymerase | New England Biolabs | Cat# M0530 |
| Phusion High-Fidelity PCR Master Mix with HR Buffer | New England Biolabs | Cat# M0531 |
| Dynabeads™ MyOne™ Streptavidin T1 | Invitrogen | Cat# 65601 |
| **Software** | | |
| MAGeCK | Li et al, 2014 | N/A |
| DESeq2 | Love et al, 2014 | N/A |
| STRING Version 11.5 | Szklarczyk et al, 2019 | N/A |
| ImageJ Software64 | Rueden et al, 2017 | N/A |
| TIDE(Tracking of Indels by Decomposition) | Brinkman et al, 2018 | N/A |
| FlowJo Version 10.8.1 | N/A | RRID:SCR_008520 |
| ScanR analysis | N/A | N/A |
| Comet score 2 | N/A | N/A |
| **Other** | | |
| *Kits, instrumentation, laboratory equipment, lab ware etc. that are critical for the experimental procedure and do not fit in any of the above categories can be listed here.* | | |
| QIAamp DNAMini Kit | Qiagen | Cat# 51306 |
| MiniElute PCR Purifcation Kit | Qiagen | Cat# 28006 |
| SE Cell line 4D Nuclefector X Kit | Lonza | Cat# V4XC-1024 |
| Subcellular Protein Fractionation Kit | Thermo Scientific | Cat# 78840 |

## Cell lines

All 2D cell lines used in this study were described previously: KB2P1.21, KP3.33 (Evers et al, 2008), DLD-1 BRCA2(−/−) (Horizon Discovery, #HD 105-007; RRID:CVCL_HD57), HEK293FT (RRID:CVCL_6911). For their culture, cell growth media were supplemented with 10% (v/v) fetal calf serum (FCS, Sigma) and 50 units/ml penicillin–streptomycin (Gibco). KB2P1.21 and KP3.33 cells were grown in Dulbecco's Modified Eagle Medium/Nutrient Mixture F-12 (DMEM/F-12; Gibco) containing 5 μg/ml Insulin (Sigma, #I0516), 5 ng/ml cholera toxin (Sigma, #C8052) and 5 ng/ml murine epidermal growth factor(EGF, Sigma, #E4127). For the growth of the DLD-1 BRCA2(−/−) cells, the medium was additionally enriched with 2 mM L-glutamine and 25 mM sodium bicarbonate. HEK293FT cells were cultured in Iscove's Modified Dulbecco's Media (IMDM, Gibco) supplemented with 10% fetal calf serum (FCS, Sigma) and 50 units/ml penicillin–streptomycin (Gibco). Tissue culture was carried out under low oxygen conditions (37 °C, 5% $CO_2$, 3% $O_2$), except for the HEK293FT cells, which were cultured under standard conditions (37 °C, 5% $CO_2$). All cell lines used in this study are of female origin, except for DLD-1 BRCA2(−/−) cells (male). Testing for mycoplasma contamination was performed regularly, twice a year.

## CRISPR/Cas9-based genetic screens

Screen 1 was performed in the KB2P and KB2P-P2 isogenic cell lines, with stable Cas9 expression. Screen 2 was performed in a clonal KB2P cell line expressing a doxycycline-inducible Cas9. In both screens, the YUSA mouse improved genome-wide knockout CRISPR library V2 (Tzelepis et al, 2016), which carried 90,230 sgRNAs, targeting 18,424 genes (5 sgRNAs/gene), was stably introduced into the cells by lentiviral transduction at the multiplicity of infection (MOI) of 1. To obtain three biological replicates for each mutagenized cell line and each condition, three independent transductions were carried out for each cell line in screen 1 (6 in total) and three for the KB2P cell line in screen 2. Each replicate in Screen 2 was separated in DMSO and 100 nM PDDX-004 inhibitor treated on day 1 of the screen. To perform the genetic screen at 500× library coverage, $45 \times 10^6$ mutagenized cells were plated in 15-cm plates in a $0.5 \times 10^6$ density. Each replicate of

each mutagenized cell line in screen 1 was kept in culture for 8 days and was passaged every 2 days. For screen 2, each replicate of the DMSO and PDDX-004 inhibitor-treated mutagenized cell line was kept in culture for 15 days and was passaged every 3 days. Cells were harvested on day 0 and at the end of each screen for genomic DNA isolation. In order to enrich only for the lentiviral integrations containing the sgRNA sequence in the gDNA samples and to reduce the PCR input, we used a DNA capture protocol adapted from Jastrzebski et al, 2016. In brief, the gDNA was digested with the NdeI and XhoI restriction enzymes and then biotinylated capture primers were hybridized to the DNA fragment containing the sgRNA sequences. The hybrids were then captured with streptavidin bids. After following an exonuclease reaction to remove any left-over capture primers, the captured gDNA fragments were amplified from genomic DNA by two rounds of PCR amplification as described previously (Sanjana et al, 2014; Xu et al, 2015). The amplified DNA libraries were sequenced in an Illumina HiSeq-2500 sequencer, and raw reads were demultiplexed and analyzed using the in-house perl script XCALIBR (https://github.com/NKI-GCF/xcalibr). After normalization, a differential test between the control (KB2P/DMSO) condition and the tested (KB2P-P2/PDDX-004) condition was performed using DESeq2 (Love et al, 2014). The output from the DESeq2 analysis contains the DESeq2 test statistic. A positive DESeq2 test statistic indicates a positive log2-transformed fold-change value, whereas a negative DESeq2 test statistic indicates a negative log2-transformed fold-change value. We sorted the output of DESeq2 for the test statistic in increasing order, including the most significantly depleted sgRNA at the top. We then used the MAGeCK Robust Rank Algorithm (Li et al, 2014) to determine for each gene whether its sgRNAs were enriched towards the top of the result list. The resulting enrichment $P$ values were corrected for multiple testing using the Benjamini–Hochberg correction, resulting in a false-discovery rate (FDR)-corrected value. As initial hits, we considered the genes with a FDR $\leq 0.1$. In order to reduce the noise level, we filtered out sgRNAs with low counts in the D0 sample and a quantile normalization was also performed for the Screen 1 results.

## STRING analysis

The protein interaction map shown in Fig. EV1D was carried out using the STRING protein–protein interaction network enrichment analysis, version 11.5. The network was formed based on evidence of experiments and databases, with high confidence (0.700). Disconnected nodes were hidden. Clusters were formed based on the KEGG pathways enriched in the network as shown in the analysis results.

## Lentiviral transductions

Lentiviral stocks, pseudotyped with the VSV-G envelope, were generated by transient transfection of HEK293FT cells, as described before (Follenzi et al, 2000). On day 0, $8 \times 10^6$ HEK293FT cells were seeded in 150-cm cell culture dishes, and on the next day, transiently transfected with lentiviral packaging plasmids and the plentiCRSIPRv2 or pLentiGuide vector containing the respective sgRNA or a non-targeting sgRNA using 2×HBS (280 nM NaCl, 100 mM HEPES, 1.5 mM $Na_2HPO_4$, pH 7.22), 2.5 M $CaCl_2$ and 0.1× TE buffer (10 mM Tris pH 8.0, 1 mM EDTA pH 8.0, diluted

1:10 with dH$_2$O). After 24 h, the virus-containing supernatant was collected. Lentiviral titers were determined using the qPCR Lentivirus Titration Kit (Applied Biological Materials), following the manufacturer's instructions. For all experiments, the amount of lentiviral supernatant used was calculated to achieve an MOI of 50, except for the transduction of the lentiviral library for which a MOI of 1 was used, as described above. Cells were incubated with lentiviral supernatants overnight in the presence of polybrene (8 mg/ml). Antibiotic selection was initiated right after transduction for cells and was carried out for 3 consecutive days. The target site modifications of the polyclonal cell pools were analyzed by TIDE analysis which is described below.

## Genome editing

For the CRISPR/Cas9-mediated targeting of the *Parg* locus two approaches had been used: (1) KB2P cells were initially transduced with the lentiviral pGS-Cas9 (Neo) construct and grown under G418 selection (500 μg/ml) for 5 days. Next, neomycin-selected cells were incubated with lentiviral supernatants of iKRUNC-Puro vectors containing the *Parg* exon 3-targeting sgRNA or a non-targeting sgRNA and exposed to 3 μg/ml puromycin for 5 days. Then, to induce sgRNA expression, puromycin-surviving cells were treated for another 5 days with 3 μg/ml doxycycline (Sigma-Aldrich). (2) KB2P, KP and DLD-1 *Brca2*−/− cells were targeted on two *Parg* exons using the Alt-R CRISPR-Cas9 system of IDT and Lonza's 4D nucleofector system. Briefly, cells were electro-porated with the ribonucleoprotein (RNP) complex consisting of the Cas9 enzyme and sgRNA sequences targeting exons 3 and 9 (mouse cell lines) or exon 7 (human cell line). The isogenic cell clones were isolated by limiting dilution.

The Cas9 expression required for the CRISPR/Cas9-based whole-genome screens was carried out by transduction with the lentiviral pGS-Cas9 (Neo) plasmid for Screen 1 or transduction with lentiviral TRE3G-Cas9 plasmid, followed by doxycycline induction, for Screen 2. The YUSA pooled sgRNA library lentiviral constructs had been propagated and transduced according to the manufacture's protocol (Tzelepis et al, 2016).

For CRISPR/Cas9-mediated targeting of *Exo1*, KB2P cells expressing the inducible Cas9 system or KB2P PARG ko cell lines (approach 2) expressing the Cas9-Neo system, were transduced with the pLentiGuidev2 vector encoding non-targeting sgRNA, Exo1-targeting sgRNA 1 or Exo1-targeting sgRNA 2. The cells were then grown under Puromycin (3 μg/ml) selection for 5 days. PCR amplicons corresponding to edited loci (amplicon primer sequences below), and gene disruption subsequently confirmed by Sanger sequencing.

For the KB2P-P1 the CRISPR sgRNA sequence of *Parg* was chosen from the GeCKo library v2 (Sanjana et al, 2014) and for the rest PARG ko cell lines were designed using the Synthego CRISPR design tool. For the *Exo1* targeting in KB2P and KB2P PARG ko cells, the sgRNA sequences were picked from the YUSA mouse library v2 (Tzelepis et al, 2016). The sgRNA sequences are listed in the Resources Table (Table 1).

## gDNA isolation, amplification, and TIDE analysis

To assess the modification rate, cells were pelleted and the genomic DNA was extracted using the QIAmp DNA mini kit (Qiagen) according to the manufacturer's protocol. Target loci were amplified

using Phusion High-Fidelity Polymerase (Thermo Scientific) using a 3-step protocol: (1) 98 °C for 30 s, (2) 30 cycles at 98 °C for 10 s, 63.8 °C for 20 s and 72 °C for 30 s, (3) 72 °C for 5 min. Reaction mix consisted of 10 µl of 2× Phusion Mastermix (Thermo Fisher), 1 µl of 20 µM forward and reverse primer, and 100 ng of DNA in 20 µl total volume. PCR products were purified using the QIAquick PCR purification kit (Qiagen) according to the manufacturer's protocol and submitted with corresponding forward primers for Sanger sequencing to confirm target modifications using the TIDE (Tracking of Indels by Decomposition) algorithm (Brinkman et al, 2014). The primers used in this PCR are listed in the Resources Table (Table 1).

## siRNA and transfections

ON-TARGETplus siRNA SMARTpools (Dharmacon) for mouse Exo1 or Fen1 and ON-TARGETplus non-targeting siRNA were transfected into the cells using Lipofectamine RNAiMAX (Invitrogen) according to the manufacturer's instructions. All experiments were carried out between 48 and 72 h post-transfection.

## RT-qPCR

In order to determine gene expression levels, RNA was extracted from cultured cells using ISOLATE II RNA Mini Kit (Bioline) and used as a template to generate cDNA with Tetro cDNA Synthesis Kit (Bioline). Quantitative RT-PCR was performed using SensiMix SYBR Low-ROX Kit (Bioline; annealing temperature—60 °C) in a Lightcycler 480 384-well plate (Roche), and analyzed using Lightcycler 480 Software v1.5 (Roche). Mouse *Hprt* was used as a housekeeping gene. The primer sequences used in this study are listed in the Resources Table (Table 1).

## Viability assay

Long- and short-term viability assays were performed in six-well platess. Cells were seeded at low density to avoid contact inhibition between the clones (KB2P/KP cell lines: 3000 cells/well; KB1P cell lines: 4000 cells/well; DLD cell lines: 2000 cells/well; Drugs were at beginning of the assay and the media was refreshed after 7 days in long-term assays. For the quantification, cells were incubated with Cell-Titer Blue (Promega) reagent and later fixed with 4% formaldehyde and stained with 0.1% crystal violet. Drug treatments: cells were grown in the continuous presence of olaparib, PDDX-004/001 or LNT1 at the indicated concentrations. The inhibitors were reconstituted in DMSO (10 mM). When a dox-inducible system was used, then cells were treated with 2 mM Doxycycline for 3 days prior to the start of the assay.

## Viability competition assays

For competition assays, 50,000 cells were seeded per 10-cm plate under treatment with PDDX-004 or PDDX-001 inhibitor at the indicated concentrations. Cells were constantly exposed to the drugs during the course of the experiments. Genomic DNA was isolated from the cells just before the start of the assay and from the remaining control or inhibitor-treated cells on day 7. TIDE analysis was performed on the samples collected on the corresponding days.

## PAR immunofluorescence analysis

Cells were seeded on Corning 96-well special optics plates (CLS3720, Sigma) 24 h prior the assay to achieve ~90% confluency. Next, cells were treated with the drugs for 1.5 h and with MMS for the last 30 min, when indicated. The drug concentration used were: 500 nM olaparib, 1 µM PDDX-004 (unless otherwise indicated), 10 µM LNT1 (824983-91-7, AxonMedChem), 2 µM emetine and 0,01% MMS. After incubation with drugs, plates were fixed with ice-cold 95% (v/v) methanol/PBS (100 µl/well) for 15 min at −20 °C. Plates were then washed twice with PBS and cells were permeabilized by adding 100 µl/well of 0.1% (v/v) Triton X-100 in PBS and incubating for 20 min at room temperature. Incubation with the primary mouse monoclonal anti-PAR (H10) antibody (MABC547, Millipore), diluted 1:4000 in PBS solution containing 5% (v/v) FBS and 0.05% (v/v) Tween-20, was carried out overnight at 4 °C. After three washes with PBS, cells were incubated for 1 h (room temperature) with polyclonal AlexaFluor488 goat anti-mouse immunoglobulins (1:1000) or AlexaFluor568 goat anti-rabbit and Hoechst (1:5000; Thermo Scientific) diluted in 5% (v/v) FBS/0.1% (v/v) Triton X-100 in PBS. PAR immunofluorescent signal was detected with a Leica SP5 confocal (Leica Microsystems). Total nuclear intensities were measured per nuclei with ImageJ software. For each well, four different areas (200 cells on average) were imaged and analyzed. Each experiment was repeated three times.

## γH2AX foci

Cells were seeded on Millicell EZ slides (#PEZGS0816, Millipore) 24 h prior the assay to achieve ~90% confluency. Cells were seeded after 48 h siRNA treatment or they were incubated with 10 µM LNT1 for 2 h, when indicated. Next, cells were washed with PBS and fixed with 4% (v/v) PFA/PBS for 20 min in RT, followed by three washes with PBS and permeabilization with 0.2% (v/v) Triton X-100 in PBS. Fixed cells were washed three times with staining buffer (5% (v/v) FBS, 5% (w/v) BSA, and 0.05% (v/v) Tween-20 in PBS) and incubated with primary antibody antiH2AX (Ser139) (1:1000, Cell Signaling, Cat#2577) in staining buffer for 2 h at RT. After three washes in staining buffer, cells were incubated with secondary antibody anti-rabbit Alexa Fluor 488 (1:500, A27034, Invitrogen) in staining buffer, followed by three last washes in staining buffer and one wash in PBS. Samples were mounted with VECTA-SHIELD Hard Set Mounting Media with DAPI (#H-1500; Vector Laboratories). Images were captured with Leica SP8 confocal (Leica Microsystems) confocal system and analyzed with ImageJ software.

## Micronuclei analysis

Following 48 h of siRNA treatment, cells were seeded on Millicell EZ slides (#PEZGS0816, Millipore). After 24 h, cells were washed with PBS and fixed with 4% (v/v) PFA/PBS for 20 min in RT. Cells were then washed three times in PBS and permeabilized for 20 min in 0.2% (v/v) Triton X-100/PBS. Then, slides were washed three times with PBS and mounted with VECTASHIELD Hard Set Mounting Media with DAPI (#H-1500; Vector Laboratories). Images were captured with Leica SP8 confocal (Leica Microsystems). Five areas were captured per well. The frequency of micronuclei-positive cells was analyzed manually in ImageJ.

## Indirect immunofluorescence

Cells were seeded onto coverslips at $6 \times 10^5$ cells per well of six-well plate. Next day, cells were treated or not with 10 µM FEN1i inhibitor for 2 h or 1 µM PDDX-004 for 30 min at 37 °C. Before fixation, cells were washed with cold PBS, pre-extracted using cold pre-extraction buffer (25 mM HEPES pH 7.4, 50 mM NaCl, 1 mM EDTA, 3 mM MgCl2, 0.3 M sucrose, 0.5% Triton X-100) 5 min on ice and fixed with cold 4% formaldehyde for 15 min. Cells were permeabilized using ice-cold methanol/acetone solution (1:1) for 5 min and PBS containing 0.5% Triton X-100 and blocked in BSA for 30 min. Cells were stained with indicated primary antibodies 1 h at RT, washed with PBS followed by incubation with secondary antibodies for 1 h at RT, after washing with PBS, DNA was stained using DAPI. Images were acquired using Olympus IX81 microscope equipped with ScanR high-content imaging platform and 20×/0.8 dry objective. Nuclei were segmented and mean intensity of signal was quantified using ScanR analysis software. Data were analyzed using FlowJo software. PCNA staining was used to determine G1 (2n, PCNA-), S phase (PCNA +) and G2 (4n, PCNA-) cells.

## Cell cycle analysis and phospho-RPA staining

Cells were cultured in 6-well plates and treated with the indicated siRNAs for 72 h or with 10 µM LTN1 for 3 h, combined with incubation 2 µM EdU incubation for the last 2 h. Cells were then trypsinized and washed with ice-cold PBS. Next, cell pellets were fixed with Fix buffer I (BD Bioscience) for 10 min in 37 °C, washed and permeabilized with Perm Buffer III (BD Bioscience) for 30 min. Cells were then washed in 10%FBS/PBS (FACS buffer) and stained with 1:400 pRPA32 (S4/S8) rabbit polyclonal antibody (A300-245A, Bethyl laboratories) for 1 h in RT. After washing with FACS buffer, samples were incubated with 1:400 AlexaFluor568 goat anti-rabbit secondary antibody and 1 mg/mL DAPI. Click reaction was carried out after another washing step using Click-iT EdU Alexa Fluor 488 Imaging Kit (Invitrogen, C10640) according to the manufacturer's instructions. Flow-cytometric analysis was performed on LSRFortessa (BD Biosciences) and data were analyzed using FlowJo Software version 10.8.1. (Tree Star Inc.).

## DNA fiber assays

DNA fiber length was measured as described previously in (Quinet et al, 2017) with a few modifications. Briefly, asynchronously growing sub-confluent cells were labeled with 25 µM thymidine analog 5-chloro-2'-deoxyuridine (CIdU) (#C6891, Sigma-Aldrich) for 20 min, washed three times with warm PBS and subsequently exposed to 250 µM of 5-iodo-2′-deoxyuridine (IdU) for 20 min. Cells were then harvested by centrifugation, washed with PBS, and ~2000 cells were spotted onto glass slides. Cells were mixed on the slide with lysis buffer (200 mM Tris-HCl pH 7.5, 50 mM EDTA, 0.5% SDS) for 3'. After this incubation step, slides were tilted at 30° to allow uniform fiber spreading. Slides were air-dried for 5', fixed at RT for 10' in Methanol-Acetic Acid (3:1), and stored at 4 °C overnight. The day after, slides were denatured 1 h in 2.5 M HCl, quickly washed in PBS, and blocked for 1 h in 10% PBS-BSA. Newly replicated DNA was stained for 2 h with primary antibodies anti-IdU (BD Biosciences; #347580) and anti-CldU (Abcam; #BU1/75 (IC1)). Slides were then washed three times in PBS followed by 1 h with the secondary antibodies anti-mouse Alexa Fluor 488 and anti-rat Alexa Fluor 594. Slides were washed three times in PBS, mounted with

fluorescence mounting medium (#S3023, Dako). Fluorescent images were acquired using the DeltaVision Elite widefield microscope (GE Healthcare Life Sciences). Multiple fields of view from each sample were imaged using the Olympus 60X/1.42, Plan Apo N, UIS2, 1-U2B933 objective and sCMOS camera at the resolution 2048 × 2048 pixels. To assess fork progression CldU + IdU track lengths of at least 120 fibers per sample were measured using the line tool in ImageJ software. Statistical analysis was carried out using GraphPad Prism 6. In the replication fork restart experiment stalling with 2 mM hydroxyurea (HU) was performed for 60 min between a 15 min CldU and IdU pulse labels. After three washes with warm PBS, IdU pulse was carried out for 45 min. Samples were then processed as described above. Replication fork restart efficiency was then analyzed by manual counting of CldU tracks only (stalled forks) and CldU + IdU tracks (restarted forks) in ImageJ.

For the detection of post-replicative gaps, cells were labeled with 25 µM CldU (15') followed by 250 µM IdU (45') with or without 10 µM LNT1. Cells were then harvested by trypsinization, washed with PBS, and pre-extracted with CSK buffer (100 mM NaCl, 10 mM MOPS pH 7, 3 mM MgCl$_2$, 300 mM sucrose and 0.5% Triton X-100 in water) for 10' at RT. Next, isolated nuclei were harvested by centrifugation and incubated for 30' at 37 °C with S1 nuclease buffer containing or not 20U/ml of S1 nuclease (#EN0321, Thermo Fisher Scientific). Nuclei were then pelleted, resuspended in PBS and spread onto microscope slides as previously described.

## Alkaline comet assays

Alkaline comet assays were performed as previously described (Breslin et al, 2006). Briefly, cells were trypsinized, counted and the same amount of cells was treated with indicated drugs. Cells were then washed with cold PBS, resuspended in cold PBS, diluted 2× with 1.2% low melting agarose (42 °C) and spread onto prepared agarose-coated slides. Slides were then lysed in lysis buffer for 1 h at 4 °C, incubated in electrophoresis buffer for 45 min at 4 °C and run in dark in the cold room at 15 V for 25 min. Slides were then neutralized using 0.4 M Tris pH 7.5 for 15 min, stained with SYBR Green (Sigma, 1:10,000) and p-phenylenediamine dihydrochloride (Thermo Fisher Scientific, 41 µg/ml) in TE buffer for 15 min at RT, washed twice with distilled water, and dried in a 37 °C incubator. Images were acquired using Leica DM6000 microscope and 10×/0.40 dry objective. Comet tails were scored using Comet Score 2.0 software with manual scoring.

## PAR immunoblotting

KB2P and KB2P PARG ko cells were seeded on 10-cm dishes 24 h prior the assay, to achieve ~90% confluency. Following treatment with 500 nM olaparib or 10 µM LNT1 for 2 h, the cells were washed with PBS, trypsinized and then lysed for 30 min in RIPA buffer supplemented with protease inhibitors (cOmplete Mini EDTA-free, Roche). The protein concentration was determined using Pierce BCA Protein Assay Kit (Thermo Scientific). SDS-Page was carried out with the Invitrogen NuPAGE SDS-PAGE Gel System (Thermo Fisher; gel: 4–12% Bis-Tris, buffer: MOPS, input: 30 µg protein), according to the manufacturer's protocol. Next, proteins were electrophoretically transferred to a nitrocellulose membrane (Biorad). Membranes were first stained with Ponceau S, imaged and then they were blocked in 5% (w/v) milk in Tris-buffered saline Tween-20 buffer (TBST; 100 mM Tris, pH 7.6, 500 mM NaCl, 0.05% (v/v) Tween-20 (blocking buffer)). Then, the membranes were incubated overnight with mouse monoclonal anti-PAR

(MABC547, Millipore), 1:1000 in blocking buffer, at 4 °C. Horseradish peroxidase (HRP)-conjugated secondary goat polyclonal anti-mouse (Dako) antibody (diluted 1:5000) incubation was performed for 1 h at room temperature in blocking buffer and signals were visualized by ECL (Pierce ECL Western Blotting Substrate, Thermo Scientific).

## PAR/MAR, γH2αX, pRPA2 (S4/8) pCHK1 (S317), Importin β immunoblotting

Cells were treated as indicated, washed twice with cold PBS and lysed in 2× Laemmli buffer lacking reducing agent (100 mM Tris-HCl pH 6.8, 4% SDS, 20% glycerol) followed by incubation at 105 °C for 5 min. Protein concentration was quantified using BCA assays (Thermo Fisher Scientific), DTT and bromophenol blue added to 0.1 M and 0.1%, respectively. Samples were resolved on Bis-Tris SDS-PAGE gels in MOPS buffer and transferred to nitrocellulose membrane (GE Healthcare). Membranes were blocked for 30 min in 1× TBS containing 0.05% Tween-20 (TBST) and 5% milk, followed by incubation with appropriate primary antibodies overnight at 4 °C. Membranes were washed and incubated with the horseradish peroxidase-conjugated secondary antibodies for 1 h at RT. After washing, signal was developed using ECL detection reagent (GE Healthcare) and chemiluminescence film (Agfa).

## PARP1 trapping

PARP1 trapping assay was adapted from (Murai et al, 2012). In brief, 24 h prior the assay, KB2P and KB2P PARG ko cells were seeded on 10-cm dishes to achieve ~90% confluency. Cells were treated with 500 nM olaparib or 10 μM LNT1 or 1 μM PDDX-004 for 2 h, with the last 30 min in the presence of 0.01% MMS, as indicated. After the treatments cells were trypsinized and subsequently lysed to isolate chromatin-bound fractions. Fractionation was performed with Subcellular Protein Fractionation Kit from Thermo Scientific (#78840, Rockford, IL, USA), following the manufacturer's instructions. Immunoblotting was carried out as described in the previous section (PAR Immunoblotting), using the rabbit polyclonal anti-PARP1 (#9542, Cell Signaling) primary antibody in dilution 1:1000 and the secondary goat polyclonal anti-rabbit immunoglobulins/HRP (Dako), diluted 1:5000.

## Data availability

This study includes no data deposited in external repositories.

## Peer review information

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

## Acknowledgements

The authors would like to thank Piet Borst (NKI, Amsterdam) and Paola Francica (University of Bern) for the critical reading of the manuscript. This work was funded by the Swiss National Science Foundation (P1BEP3_195482 to CA and 320030M_219453 to JJ and SR), the European Research Council (ERC-2019-AdG-883877 to SR), the Swiss Cancer League (KFS-5519-02-2022 to SR), the Department of Defense [W81XWH-22-1-0557 to SR] and the Wilhelm-Sander Foundation (no. 2019.069.1 to SR). Research at the NKI is supported by institutional grants of the Dutch Cancer Society and the Dutch Ministry of Health, Welfare, and Sport. This work is part of the Oncode Institute which is partly financed by the Dutch Cancer Society. This work was additionally funded by the Dutch Cancer Society (KWF 2017-61169 to JJ and KWF 2020-12894 to JJ), as well as the Czech Science Foundation (no. 22-00885S to HH).

## Author contributions

**Christina Andronikou**: Conceptualization; Resources; Data curation; Software; Formal analysis; Funding acquisition; Validation; Investigation; Visualization; Methodology; Writing—original draft; Project administration; Writing—review and editing. **Kamila Burdova**: Conceptualization; Investigation; Methodology. **Diego Dibitetto**: Investigation; Methodology. **Cor Lieftink**: Formal analysis; Visualization. **Elke Malzer**: Resources. **Hendrik J Kuiken**: Resources. **Ewa Gogola**: Resources. **Arnab Ray Chaudhuri**: Resources. **Roderick L Beijersbergen**: Resources; Methodology. **Hana Hanzlikova**: Supervision; Funding acquisition; Methodology. **Jos Jonkers**: Supervision; Funding acquisition; Project administration; Writing—review and editing. **Sven Rottenberg**: Conceptualization; Supervision; Funding acquisition; Project administration; Writing—review and editing.

## Disclosure and competing interests statement

The authors declare no competing interests.

# Expanded View Figures

**Figure EV1.** *Parg*⁻/⁻ **and PDDX-004-treated KB2P cells show increased PAR levels and are resistant to the PARPi olaparib.**

(A) Sanger sequencing fragments of KB2P-NT, KB2P-P1 and KB2P-P2, corresponding to the targeted sgRNA sequences (sgRNA3-1, sgRNA3-2, sgRNA9) and the subsequent flanking regions, which confirm the successful introduction of deleterious mutations in the mouse *Parg*. (B) CTB-based viability quantification and representative images of the long-term clonogenic assay of the KB2P-NT, KB2P-P1 and KB2P-P2, in the absence or presence of 50 and 100 nM of olaparib. The data are representative for three independent experiments, shown as mean ± SD of replicates, two-tailed *t* test, **$P < 0.01$, ***$P < 0.001$. (C) Intensity quantification and representative images of PAR immunofluorescence in KB2P-NT, KB2P-P1 and KB2P-P2 cells, 30 min following the treatment with 0.01% MMS, with or without 1.5 h pre-treatment of 500 nM olaparib. The data are representative of three independent experiments and are shown as mean ± SD of $n = 3$, two-tailed *t* test, ***$P < 0.001$, ****$P < 0.0001$. Scale bar 50 μm. (D) STRING analysis network of the 221 gene hits in the screen analysis. The 5 main gene clusters are indicated and the related KEGG pathways are annotated. (E) Table showing the pathways implicated in Cluster 1 from (D), which consists of the top 12 gene hits. (F) CTB-based viability quantification and representative images of the long-term clonogenic assay of the KB2P cells in the absence or presence of 50 and 100 nM of olaparib combined with 0, 100 and 500 nM PDDX-004. Data are representative for three independent experiments, shown as mean ± SD of replicates, two-tailed *t* test, **$P < 0.01$. (G) Intensity quantification and representative images of PAR immunofluorescence in the KB2P cells after treatment with 0.01% MMS, with or without 1.5 h pre-treatment with 1 μM PDDX-004. Data are shown as mean ± SD of triplicates, two-tailed *t* test, ****$P < 0.0001$. Scale bar 100 μm.

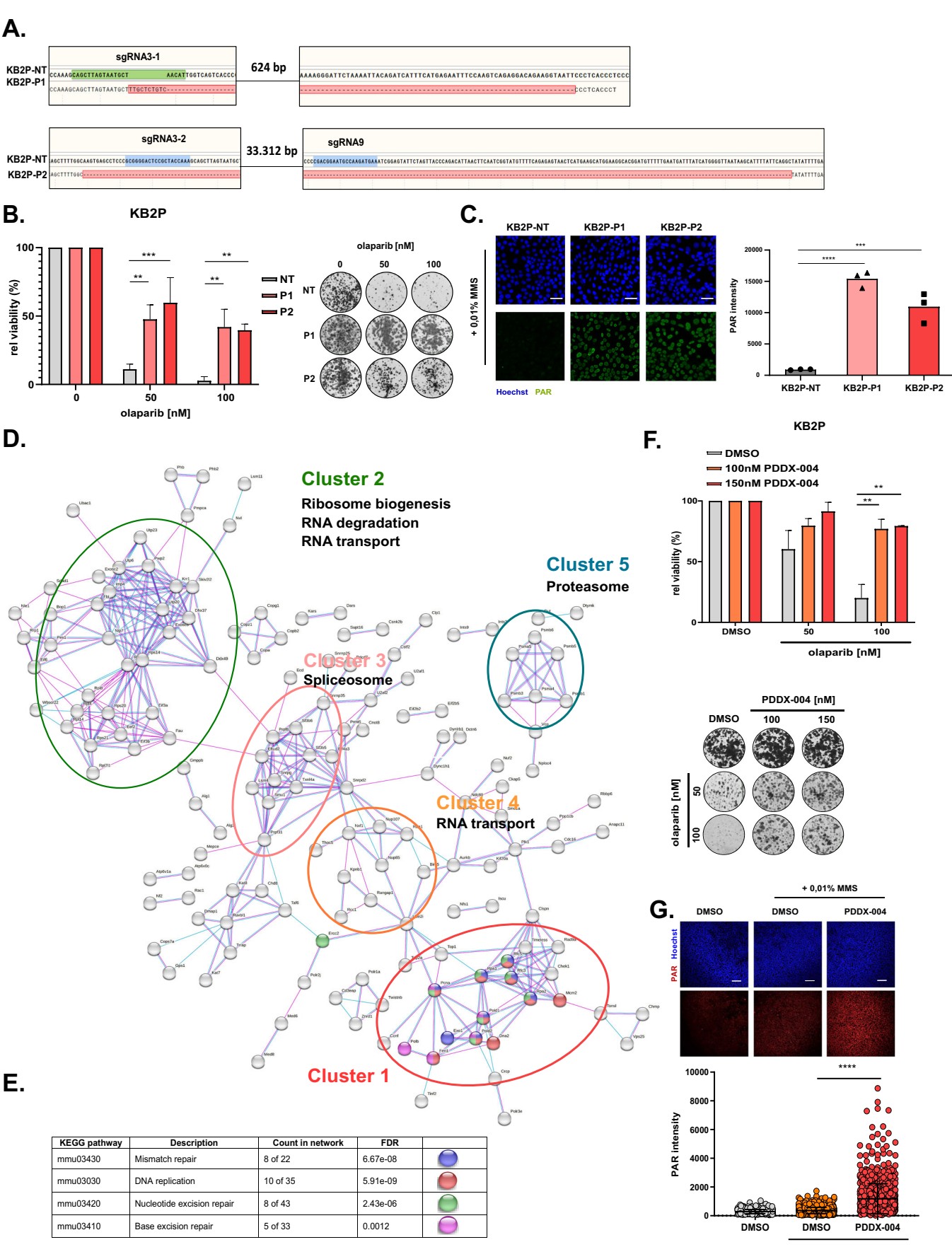

A.

B. KB2P

C.

D.

Cluster 2
Ribosome biogenesis
RNA degradation
RNA transport

Cluster 5
Proteasome

Cluster 3
Spliceosome

Cluster 4
RNA transport

Cluster 1

E.

| KEGG pathway | Description | Count in network | FDR | |
|---|---|---|---|---|
| mmu03430 | Mismatch repair | 8 of 22 | 6.67e-08 | |
| mmu03030 | DNA replication | 10 of 35 | 5.91e-09 | |
| mmu03420 | Nucleotide excision repair | 8 of 43 | 2.43e-06 | |
| mmu03410 | Base excision repair | 5 of 33 | 0.0012 | |

F. KB2P

G.

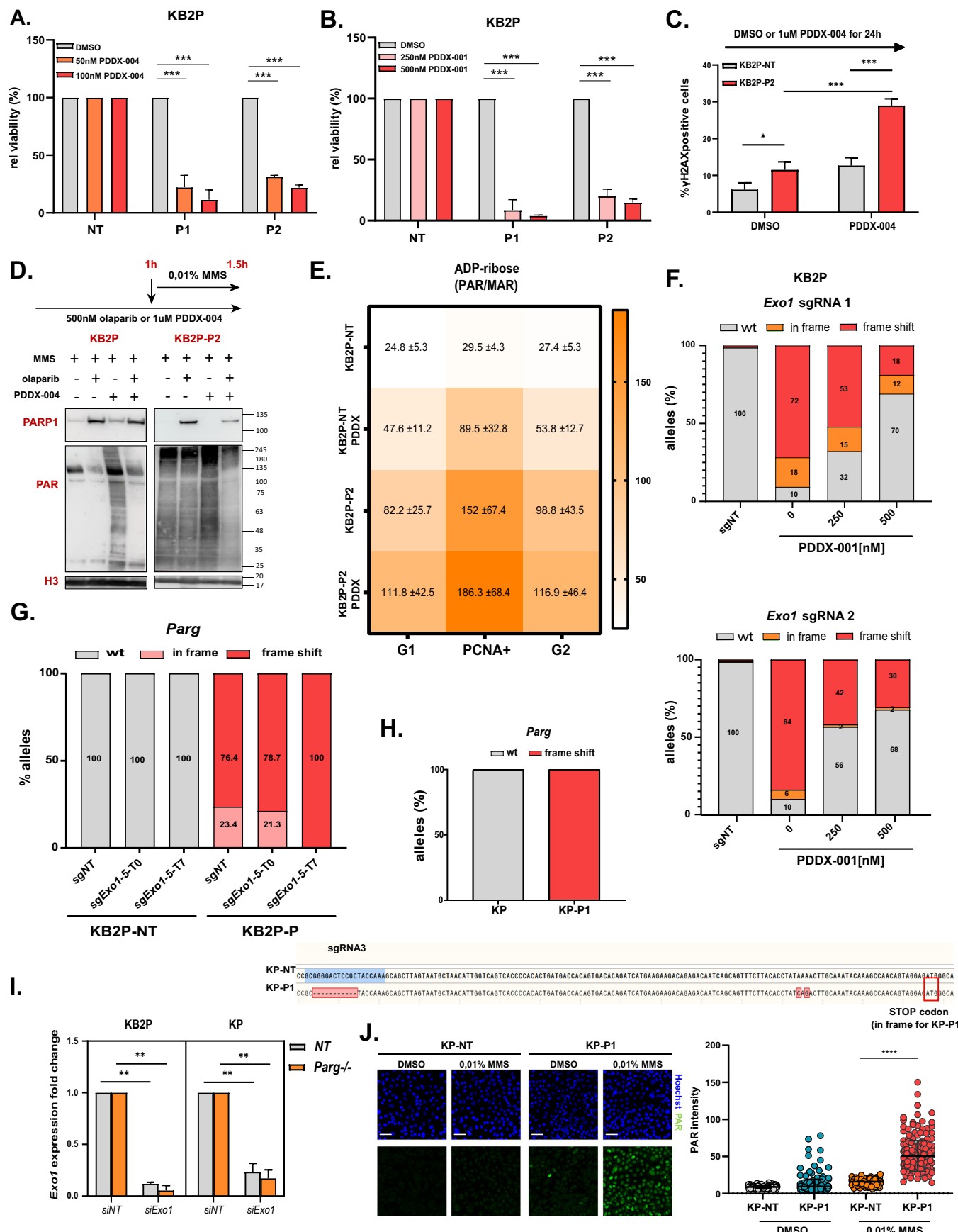

◀ **Figure EV2. PDDX inhibitor treatment and *Exo1* depletion are lethal for *Parg⁻ᐟ⁻;Brca2⁻ᐟ⁻;p53⁻ᐟ⁻* cells.**

(A) CTB-based quantification of the long-term viability assay of the KB2P-NT, KB2P-P1 and KB2P-P2 cells, in the absence or presence of 50 and 100 nM of PDDX-004. Data are representative for three independent experiments, shown as mean ± SD of replicates, two-tailed $t$ test ***$P < 0.001$. (B) CTB-based quantification of the long-term viability assay of the KB2P-NT, KB2P-P1 and KB2P-P2 cells, in the absence or presence of 250 and 500 nM PDDX-001. The data are representative for three independent experiments, shown as mean ± SD of replicates, two-tailed $t$ test ***$P < 0.001$. (C) IF analysis of γH2AX foci in KB2P-NT and KB2P-P2 cells, after 24 h treatment with 1 μM of PDDX-004. Data are representative of three independent experiments shown as mean ± SD of $n = 3$, two-tailed $t$ test *$P < 0.05$, ***$P < 0.001$. (D) Immunoblot analysis of PARP1, PAR and Histone 3 (H3) in chromatin-bound fractions of KB2P-NT and KB2P-P2 cells upon 1.5 h treatment with DMSO, 500 nM olaparib or 1 μM PDDX-004, with 0,01% MMS treatment added for the last 30 min. (E) Heatmap representing the ScanR quantification of median nuclear intensities of anti-ADP-ribose (PAR/MAR) immunofluorescence of KB2P-NT and KB2P-P2 cells following 30 min incubation with or without 1 μM PDDX-004. Data are shown as mean ± SD of $n = 3$. (F) Allelic modification rates of *Exo1* in KB2P cells upon targeting with two independent sgRNA sequences and following treatment with 0, 250 and 500 nM PDDX-001 for 1 week. (G) Allelic modification rates of *Parg* in KB2P-NT and KB2P-P (polyclonal *Parg⁻ᐟ⁻*) cells upon sgRNA-mediated targeting of *Exo1*, at day 1 after puromycin selection (T0) or after 7 days in culture (T7)- corresponding to Fig. 2B. Evaluated by TIDE analysis. (H) Allelic modification rates of *Parg* in the sgParg3-targeted locus of KP-NT and KP-P1 (up) and Sanger sequencing fragments of KP-NT and KP-P1 containing the the targeted sgRNA3 sequence in mouse *Parg* and the formed in-frame STOP codon sequence (down). (I) RT-qPCR analysis of *Exo1* expression in KB2P-NT, KB2P-P2, KP-NT and KP-P1 cells. Data are shown as mean ± SD of $n = 3$, two-tailed $t$ test, **$P < 0.01$. (J) Intensity quantification and representative images of PAR immunofluorescence in the KP-NT and KP-P1 cells 30 min after treatment with 0.01% MMS. Data are shown as mean ± SD of triplicates, two-tailed $t$ test, ****$P < 0.0001$. Scale bar 50 μm.

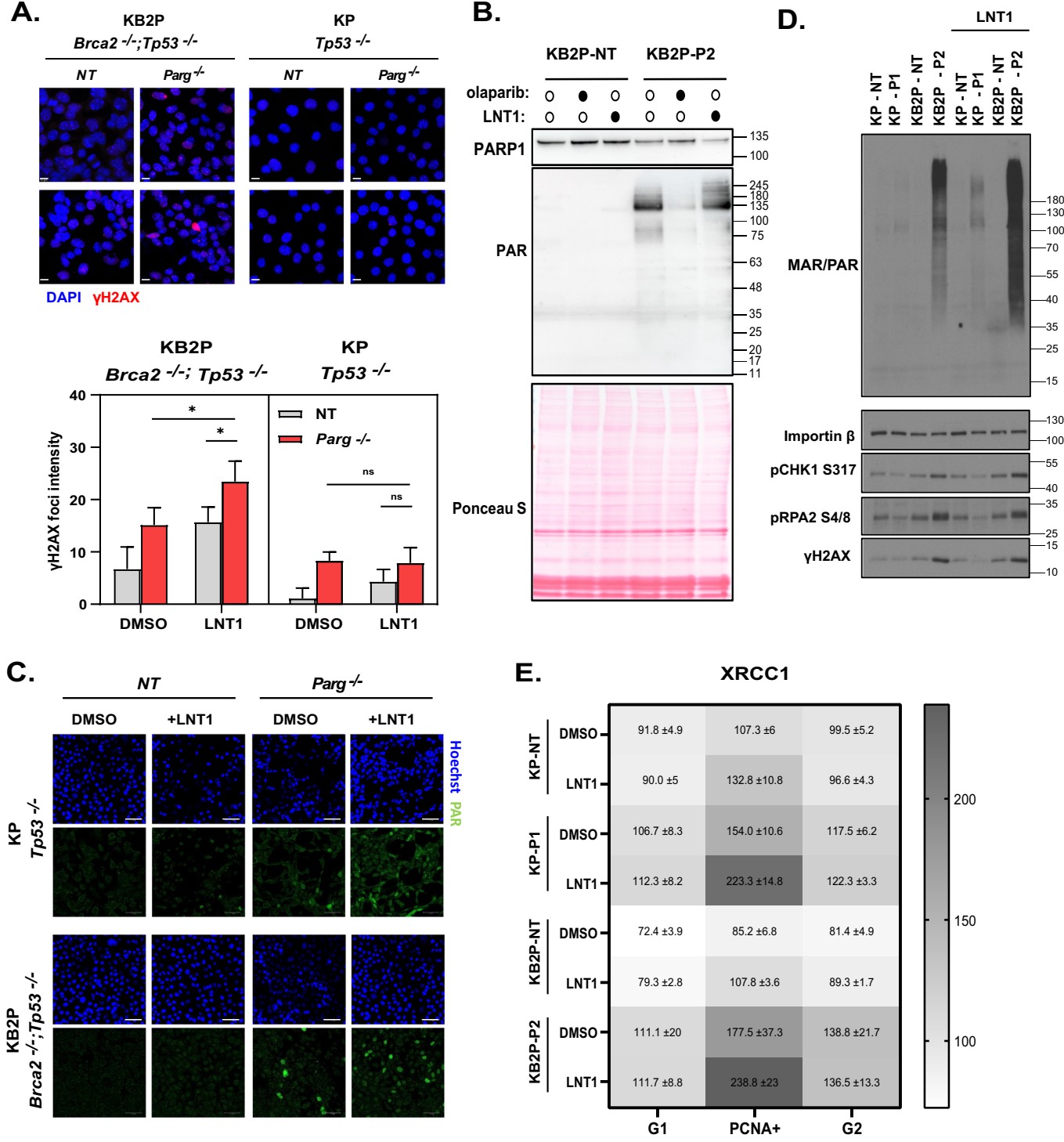

**Figure EV3. LNT1 treatment results in increased DNA damage but not increased PARP1 trapping in *Parg⁻/⁻;Brca2⁻/⁻;p53⁻/⁻* cells.**

(A) IF analysis and representative images of γH2AX foci in KB2P-NT, KB2P-P2, KP-NT and KP-P1 cells, before (upper panels) and after (lower panels) 2 h treatment with 10 µM of LNT1. Data representative of three independent experiments and are shown as mean ± SD of n = 3, two-tailed t test ns, non-significant, *P < 0.05. Positive cells >= 10 foci. Scale bar 100 µm. (B) Immunoblot analysis of PARP1 and PAR in chromatin-bound fractions of KB2P-NT and KB2P-P2 cells upon 2 h treatment with DMSO, 1 µM olaparib or 10 µM LNT1. (C) Representative images of PAR immunofluorescence in KB2P-NT, KB2P-P2, KP-NT and KP-P1 cells, 2 h after treatment with 10 µM LNT1. Scale bar 50 µm. Related to Fig. 4B. (D) Immunoblot analysis MAR/PAR, pCHK1 S317, pRPA2 S4/8 and γH2aX in lysates of KB2P-NT, KB2P-P2, KP-NT and KP-P1 cells after 30 min treatment using DMSO or 10 µM LNT1. (E) Heatmap representing the ScanR quantification of median nuclear intensities of XRCC1 immunofluorescence of KB2P-NT, KB2P-P2, KP-NT and KP-P1 cells following 30 min incubation with 10 µM LNT1. Data are shown as mean ± SD of n = 3. Source data are available online for this figure.

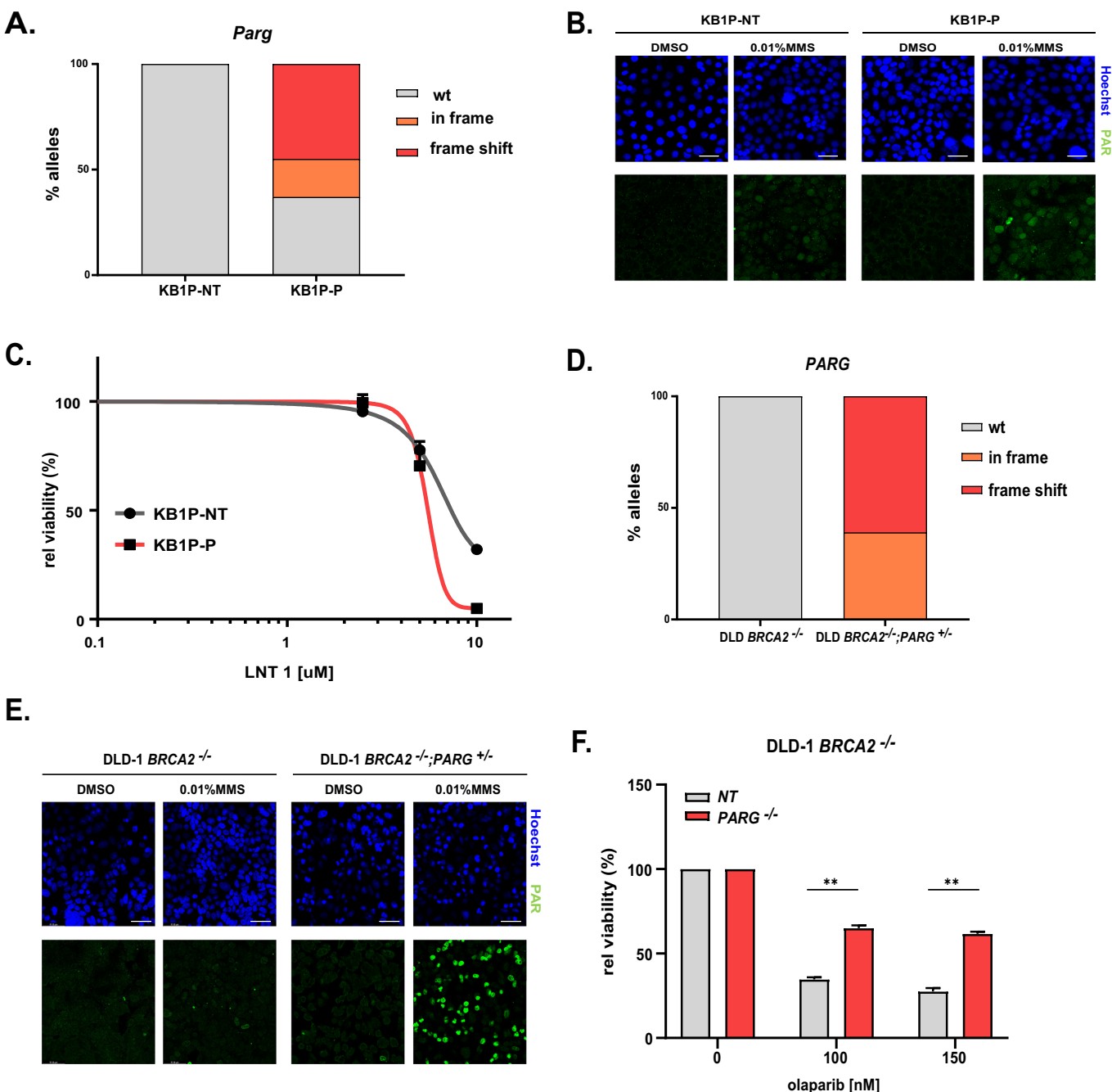

**Figure EV4. PARG-deficient DLD-1 BRCA2(-/-) and KB1P cells show increased PARylation and sensitization to LNT1 treatment.**

(A) Allelic modification rates of mouse *Parg* in the *sgParg3* locus of KB1P and KB1P cells in which *Parg* was depleted. (B) PAR immunofluorescence of KB1P and KB1P PARG-deficient cells following 30 min of treatment with 0.01% MMS. Scale bar 50 μm. (C) CTB-based viability quantification and representative images of the long-term clonogenic assay of the KB1P-NT, KB1P PARG ko cells in the presence of 0, 2.5, 5 and 10 μM of LNT1. Data are representative for three independent experiments, shown as mean ± SD of replicates. (D) Allelic modification rates of human *PARG* in the *sgParg7* locus of DLD-1 *BRCA2−/−* and DLD-1 *BRCA2−/−;PARG−/−* cells. (E) PAR immunofluorescence of DLD-1 *BRCA2−/−* and DLD-1 *BRCA2−/−;PARG−/−* cells following 30 min of treatment with 0.01% MMS. Scale bar 50 μm. (F) CTB-based viability quantification of the long-term clonogenic assay of the DLD-1 *BRCA2 −/−* and DLD-1 *BRCA2 −/−;PARG−/−* cells in the presence of 0, 100 and 150 nM of olaparib. Data are shown as mean ± SD of replicates, two-tailed *t* test **$P < 0.01$.

