## [Peer Review File · The EMBO Journal]

PARG-deficient tumor cells have an increased dependence on EXO1/FEN1-mediated DNA repair

Christina Andronikou, Kamila Burdova, Diego Dibitetto, Cor Lieftink, Elke Malzer, Hendrik Kuiken, Ewa Gogola, Arnab Ray Chaudhuri, Roderick Beijersbergen, Hana Hanzlikova, Jos Jonkers, and Sven Rottenberg

Corresponding author(s): Sven Rottenberg (sven.rottenberg@unibe.ch) , Jos Jonkers (j.jonkers@nki.nl)

Review Timeline:

Submission Date:	26th Jun 23
Editorial Decision:	28th Jul 23
Revision Received:	11th Dec 23
Editorial Decision:	8th Jan 24
Revision Received:	22nd Jan 24
Accepted:	23rd Jan 24

Editor: Hartmut Vodermaier

Transaction Report:

Prof. Sven Rottenberg
University of Bern
Pathobiology and Bern Center of Precision Medicine
Laenggassstrasse 122
Bern 3012
Switzerland

28th Jul 2023

Re: EMBOJ-2023-114851
PARG-deficient tumor cells have an increased dependence on EXO1/FEN1-mediated DNA repair

Dear Sven,

Thank you again for submitting your study on synthetic lethality with PARG deficiency to The EMBO Journal. We have now received the reviews of three expert referees, copied below for your information. All referees appreciate the timeliness of the study and potential interest of the work, but also raises a number of important questions that would need to be satisfactorily answered prior to acceptance. Should you be able to adequately address these concerns, we would be happy to consider a revised manuscript further for publication.

Since it is our policy to consider only a single round of major revision and therefore important to fully answer to all comments at the time of resubmission, I would invite you to get back to me with a tentative response letter/revision plan already during the early stages of the revision work. On the basis of this response, we could then further discuss the revision requirements and how to best address the key concerns e.g. via a follow-up video call. I should add that we could also offer extension of the default three-months revision period if needed, with our 'scooping protection' (meaning that competing work appearing elsewhere in the meantime will not affect our considerations of your study) remaining of course valid also throughout this extension.

Detailed information on preparing, formatting and uploading a revised manuscript can be found below and in our Guide to Authors. Thank you again for the opportunity to consider this work for The EMBO Journal, and I look forward to hearing from you in due time.

With kind regards,

Hartmut

9) Digital image enhancement is acceptable practice, as long as it accurately represents the original data and conforms to community standards. If a figure has been subjected to significant electronic manipulation, this must be clearly noted in the figure legend and/or the 'Materials and Methods' section. The editors reserve the right to request original versions of figures and the original images that were used to assemble the figure. Finally, we generally encourage uploading of numerical as well as gel/blot image source data; for details see: embopress.org/page/journal/14602075/authorguide#sourcedata

At EMBO Press, we ask authors to provide source data for the main manuscript figures. Our source data coordinator will contact you to discuss which figure panels we would need source data for and will also provide you with helpful tips on how to upload and organize the files.

In the interest of ensuring the conceptual advance provided by the work, we recommend submitting a revision within 3 months (26th Oct 2023). Please discuss the revision progress ahead of this time with the editor if you require more time to complete the revisions. Use the link below to submit your revision:

Link Not Available

Referee #1:

PARP inhibitors have been successfully used to treat HR-deficient cancers; nevertheless, the patients can eventually develop resistance. The authors and others have previously reported that PARG loss is an important cause of PARPi resistance in murine Brca2-deficient breast cancer models and in BRCA2 mutated human tumors. Identifying the genetic vulnerabilities for PARG- and BRCA2-deficient tumors (TNBC) can therefore provide new approaches for PARPi-resistant tumors. This stimulated the authors to use whole genome CRISPR/Cas9 screens to identify synthetic lethal targets for PARG impairment in their murine model of Brca2 deficiency. EXO1 and FEN1 were among the top hits in the CRISPR dropout screen in KB2P tumor cells. Knockdown experiments or chemical inhibition using a dual inhibitor of EXO1 and FEN1 exaggerated replication defects in PARG deleted, KB2P (Brca2, p53, and PARG triple mutant) cells. The authors proposed that FEN1/EXO1 depletion results in replication defects, that are further exacerbated in the context of PARG deficiency, and this contributes to the toxicity for HR-deficient cells.

This is an interesting study to the DNA repair community that has implications for overcoming PARPi resistance in BRCA mutated cancers. However, some of the data is insufficiently developed to support the conclusions. The authors should provide more convincing evidence to better dissect EXO1/FEN1's role in replication fork progression, SSBR, and Okazaki fragment maturation in the PARG; BRCA2-deficient model.

Major points:

1. The authors describe off-target effect of PARG inhibitors: PDDX-004/ PDDX-001 inhibitors. While they make considerable

effort to perform PARG genetic deletion experiments in parallel to support their conclusions, more description of the PARGi offtarget effects would be helpful.

2. Figure 3A, 3B, the authors observed significant increases in γ H2AX- and micronuclei-positive, while also observing some pan-nuclear γ H2AX signal in KB2P-P2 cells following Exo1 depletion. They conclude this is linked to replication stress. The authors need to provide more direct evidence to support the conclusion that EXO1 knockdown enhances replication stress in KB2P-P2 cells. For example, showing S-phase associated γ H2AX signal, pATR, and pRPA2. Metaphase chromosome experiments could also support this conclusion by showing sister chromatid breaks.

3. The authors should use S1 nuclease treatment during DNA fiber experiments to support the statement that EXO1/FEN1 depletion increases unresolved ssDNA gaps. This would require an observation that S1 treatment preferentially shortens fibers in PARG deleted cells.

4. The authors used LNT1 inhibitor to target both FEN1 and EXO1. Insufficient description of this inhibitor is provided, including whether it has additional targets.

5. Regarding to the PARP trapping assay and PARylation immunoblot, the authors may need to deplete FEN1 or EXO1 individually to confirm their separate contributions. Additionally, in Figure EV 3D, LNT1 treatment seems not to induce pRPA2 S4/8 or γ H2AX (Lane 4 vs 8).

6. Figure 5C, EXO1 and FEN1 depletion showed a synergistic effect on cell viability in KB2P-P2 cells. In this context, the authors should explore the potential distinct functions between them.

Minor points:

1. According to the sanger sequencing, both KB2P-P1 and KB2P-P2 show large deletions(Figure EV1A). The authors should include a larger sequence region to show the whole deletion region and the flanking sequence. In Figure EV2E, the author should indicate the stop codon of the frameshift mutation and if any essential domain of PARG is damaged by the mutation.

2. The authors should indicate the treatment of Olaparib and MMC in figure EV1C.

Referee #2:

In the manuscript provided, authors present work showing that EXO1 or FEN1 inhibition causes synthetic lethality in cells with both BRCA2 and PARG loss. The same authors previously showed that loss of PARG causes PARP inhibitor resistance in BRCA2 mutant mice - this makes the case for identifying the weaknesses in tumor cells caused by both BRCA2 and PARG interesting (although, as the authors acknowledge, PARG defects have not been seen in patients with PARP inhibitor resistance).

The data is well presented and experiments appear to be well carried out and analyzed. The flow of the manuscript is clear and overall the manuscript makes a clear contribution to the literature and understanding of the subject area (of that there is no doubt). My only suggestions are:

1. The analysis is very focused on mouse cells, with only a small number of experiments in human cells with a BRCA2 mutation (DLD1, which has an engineered BRCA2 mutation, not a naturally occurring one). I could not see any validation of the key synthetic lethal phenotypes in human tumor cells with endogenous BRCA2 LoF mutations such as Capan 1 and PEO1. There are also HR proficient derivatives of these that could also be used to prove the phenotypes seen are dependent or independent of BRCA2 status as well. These experiments are important to eliminate the possibility that these are mainly mouse phenotypes. This is also important as there looks to be some inconsistency between what is reported in human tumor cells (see below) and what happens in engineered mouse cells.

2. Is there a reason why Mengwasser et al., 2019, Molecular Cell 73, 885-899 (Elledge) is not mentioned or taken into consideration? This manuscript shows rather convincingly that FEN1 is synthetic lethal with BRCA2 in human cell lines. How can the authors explain the observation in Figure 5B where it is only the combined loss of BRCA2 and PARG that causes sensitivity to FEN1 siRNA (with no sensitivity when only BRCA2 and p53 are defective)? Is this a mouse specific phenotype and not seen in human cells (there is no human expt using FEN1 siRNA only using a not very potent FEN1/EXO1 inhibitor in Figure 5D).

3. It is more common to show dose response curves for the small molecule expts (throughout). In general, the experiments using the low potency FEN1/EXO1 inhibitor are not as convincing as those where genetic inhibition is used.

Referee #3:

In the manuscript entitled 'PARG-deficient tumor cells have an increased dependence on EXO/FEN1-mediated DNA repair'

Andronikou et al. performed two CRISPR-Cas9 whole genome drop-out screen to identify genes essential in the survival of PARG;BRCA2;p53-deficient mice tumour cells using both genetic depletion of PARG and chemical PARG inhibition. Their screens identified various genes essential for the survival of PARG;BRCA2;p53-deficient cells. In particular, they revealed that loss of EXO1 and FEN1 is synthetically lethal with loss of PARG activity in these BRCA2;p53-deficient cells. Furthermore, they demonstrate that PARG;BRCA2-deficient cells exhibit increased replication stress and PAR levels which is further exacerbated upon EXO1 and FEN1 inhibition, resulting in cell death. Lastly, they show that EXO1/FEN1 inhibition (using LNT1 inhibitor) was also toxic to human PARG deficient tumor cells, thus identifying the two enzymes as potential drug target for tackling PARPi resistant tumors with PARG downregulation.

Overall, we found the experiments to be well designed and executed. The findings described here have important implications for future development of PARG inhibitors and we believe that the manuscript should be published in the EMBO journal after minor revisions. We just have a few clarifications to suggest.

Main comments:

1. PARG itself was identified as a hit in the CRISPR drop-out screen performed with chemical PARG inhibition. The authors also show that PARG;BRCA2-deficient cells are sensitive to both PDDX-001 and PDDX-004. This is surprising and indicates that both PARG inhibitors have off-target effects upon PARG loss, as stated by the authors. Could the authors comment further on what these off-target effects could possibly be and what implications do these observations have for the future clinical use of PARGi?

2. In Figure 4D, the nucleus size appear significantly smaller in the following conditions compared to rest of the images : LNT1+EME treatment in KBP2-NT and KBP2-P2, and EME treatment in KB2P-P2. Could the authors double check that the same magnification was used for all the representative images chosen for the figure? If so, is this smaller nucleus a representative feature of the specific condition tested or an artefact of the area chosen for imaging? If the authors think this is representative of the whole well, could they comment on why they think these specific conditions lead to a reduction in nucleus size?

3. Identifying the substrates of PAR triggered by these unresolved DNA lesions is beyond the scope of this paper. However, could the authors comment on what these substrates might possibly be in the discussion? Do the authors believe that it is primarily PARP1 auto-modification as suggested by their schematic (Fig 5E), if so this should be made clear in the text that refers to Figure 5E.

4. In the 'PAR/EXO1 immunofluorescence' analysis paragraph in the Methods section, the authors describe using the polyclonal anti-EXO1 antibody for an immunofluorescence experiment. However, as far as we could see, no experiment looking at EXO1 signal by IF can be found in the results. Could the authors update the methods section accordingly?

Additional comments:

1. On page 3, line 61, Sharifi et al. 2013 (10.1038/emboj.2013.51) should be cited at the end of this sentence.
2. On page 5, line 103 and page 18, line 451, the authors could add that a newly developed PARGi have recently entered clinical trials (IDEAYA bioscience), making the results described in this study even more relevant for future clinical applications.
3. Could the authors provide molecular weights for their immunoblotting figures (EV3B and EV3D)? This is especially important for the PAR blots.
4. Representative image for KB2P-P2 in Figure EV1C is blurry in both the PAR and Hoechst channel.
5. Could a higher magnification be used for the representative images of Figure EV1G. It is currently really hard to even distinguish individual cells with the magnification chosen.
6. On page 15, line 381, "in" or equivalent phrasing should be added after "understanding"

POINT-BY-POINT RESPONSE TO REFEREES' COMMENTS***Reviewers comments in black and our responses in red***

Referee #1:

PARP inhibitors have been successfully used to treat HR-deficient cancers; nevertheless, the patients can eventually develop resistance. The authors and others have previously reported that PARG loss is an important cause of PARPi resistance in murine *Brca2*-deficient breast cancer models and in *BRCA2* mutated human tPARumors. Identifying the genetic vulnerabilities for PARG- and *BRCA2*-deficient tumors (TNBC) can therefore provide new approaches for PARPi-resistant tumors. This stimulated the authors to use whole genome CRISPR/Cas9 screens to identify synthetic lethal targets for PARG impairment in their murine model of *Brca2* deficiency. EXO1 and FEN1 were among the top hits in the CRISPR dropout screen in KB2P tumor cells. Knockdown experiments or chemical inhibition using a dual inhibitor of EXO1 and FEN1 exaggerated replication defects in PARG deleted, KB2P (*Brca2*, p53, and PARG triple mutant) cells. The authors proposed that FEN1/EXO1 depletion results in replication defects that are further exacerbated in the context of PARG deficiency, and this contributes to the toxicity for HR-deficient cells.

This is an interesting study to the DNA repair community that has implications for overcoming PARPi resistance in *BRCA* mutated cancers. However, some of the data is insufficiently developed to support the conclusions. The authors should provide more convincing evidence to better dissect EXO1/FEN1's role in replication fork progression, SSBR, and Okazaki fragment maturation in the PARG; *BRCA2*-deficient model.

We appreciate that the reviewer finds our work interesting and useful for the field. We also thank him/her for the critical feedback to further improve our manuscript.

Major points:

1. The authors describe an off-target effect of PARG inhibitors: PDDX-004/ PDDX-001 inhibitors. While they make considerable effort to perform PARG genetic deletion experiments in parallel to support their conclusions, more description of the PARGi off target effects would be helpful.

We thank the reviewer for this comment, and we fully agree that understanding the PARG inhibitors' off-target effects is important for the field. Currently, we do not know what the off-target is/are, and elucidating the precise mechanism is beyond the scope of this manuscript. Still, we feel that it is important for the field to know about this off-target effect, since many groups are using these inhibitors. Thus far, we have not seen an experiment in the literature in which PDDX-004/ PDDX-001 were tested in PARG-deficient cells, which we think is an important control experiment for the specificity of the inhibitor. We present these data in Fig. EV2A and EV2B. Regarding the phenotypic characteristics of the off-target effect, we now show that the PDDX-mediated toxicity is characterised by an underlying increase in genomic instability (Figure EV2C) which is not an outcome of PARP1 trapping (Figure EV2D), but it's rather accompanied by elevated ADP-ribosylation levels (Figure EV2E). We now discuss these observations in lines 171-195 of the revised manuscript as follows:

“ However, in addition to *Exo1*, sgRNAs targeting *Parg* were also depleted, indicating a specific PDDX inhibitor toxicity for PARG-deficient KB2P cells. (Figure 1E). Indeed, both KB2P-P1 and -P2 cells were highly sensitive to treatment with the PDDX-004 inhibitor (Figure EV2A), paralleled by a similar response to the structurally similar compound, PDDX-001 (Figure EV2B), as evidenced in long-term viability assays. In contrast, the PARG-proficient KB2P-NT cells displayed resilience to these treatments. Corroborating these viability data, PDDX-004 treatment led to increased DNA damage specifically in PARG-deficient KB2P-P2 cells but not PARG-proficient KB2P-NT cells. This was evidenced by a notable accumulation of γ H2AX foci under these treatment conditions (Figure EV2C). To discern whether the cytotoxicity of PDDX compounds involves a PARP1 trapping mechanism akin to that observed with PARP inhibitors, we employed the trapping assay as described by Murai et al, (2012). Contrary to expectations, PDDX-004 treatment did not lead to an increase in the chromatin-bound PARP1 in KB2P-NT and KB2P-P2 cells after induction of DNA damage with MMS. Instead, it rather led to an additional increase in the ADP-ribosylation levels of KB2P-P2 cells (Figure EV2D). Subsequently, we set out to investigate in more detail how PDDX-004 treatment modulates ADP-ribosylation levels of KB2P-NT and KB2P-P2 cells under unperturbed conditions. Using high-content indirect immunofluorescence with PCNA staining to determine G1 (2n, PCNA-), S phase (PCNA+), and G2 (4n, PCNA-) cells, we observed that PDDX-004 treatment not only significantly increased ADP-ribosylation (PAR/MAR) levels in KB2P-NT cells (two-way ANOVA; $p < 0.0001$) but also further augmented the already heightened PAR/MAR levels of KB2P-P2 cells (two-way ANOVA; $p = 0.0010$) (Figure EV2E). Given that KB2P-P2 clonal cells lack all catalytically active isoforms due to their large *Parg* deletion (Figure EV1A), these results hint at the possibility that their increased sensitivity to PDDX may stem from off-target inhibition of other ADP-ribose hydrolases.”

Macrodomain-containing ADP-ribose hydrolases like MacroD1, MacroD2 and TARG1 are interesting candidate PDDX inhibitor off-targets, due to their similar catalytic domain as PARG. However, siRNA-mediated depletion of these hydrolases did not result in a viability or ADP-ribosylation effect equally strong as the PDDX inhibitor treatment on PARG-deficient KB2P cells. It is possible that siRNA-mediated depletion of the hydrolases is not as strong as the PDDX off-target effect or that the combinational depletion of multiple off-targets is underlying this effect. A more unbiased, mass-spectrometry based-approach, may be required for the off-target elucidation, but we believe that this is beyond the scope of the current manuscript.

2. Figure 3A, 3B, the authors observed significant increases in γ H2AX- and micronuclei-positive, while also observing some pan-nuclear γ H2AX signal in KB2P-P2 cells following *Exo1* depletion. They conclude this is linked to replication stress. The authors need to provide more direct evidence to support the conclusion that EXO1 knockdown enhances replication stress in KB2P-P2 cells. For example, showing S-phase associated γ H2AX signal, pATR, and pRPA2. Metaphase chromosome experiments could also support this conclusion by showing sister chromatid breaks.

We thank the reviewer for his/her wish to further confirm the enhanced replication stress. We already presented S phase-specific pRPA2 in Figure 3D. To further corroborate the enhanced replication stress, we performed analysis of the S phase-associated γ H2AX levels, following the treatment with the EXO1/FEN1 inhibitor, which is now shown in Figure 4F. Our results validate that the unresolved PAR-regulated ssDNA gaps lead to accumulation of DSBs and

replication stress, as indicated by the elevated γ H2AX levels during S phase. We now discuss these observations in lines 343-346 of the revised manuscript as follows:

“Consistent with our earlier observations, LNT1 treatment resulted in a significant increase in γ H2AX levels specifically during S phase, in KB2-P2 cells (student’s t-test; $p=0.0164$), paralleling the pattern observed in their ADP-ribosylation levels under these conditions (Figure 4F).”

3. The authors should use S1 nuclease treatment during DNA fiber experiments to support the statement that EXO1/FEN1 depletion increases unresolved ssDNA gaps. This would require an observation that S1 treatment preferentially shortens fibers in PARG deleted cells.

We agree with the reviewer that this is a useful experiment to further support our findings. We have now performed fiber analysis following EXO1/FEN1 inhibition, with or without S1 nuclease treatment. Our results, presented in Figure 4E, confirm that indeed EXO1/FEN1 inhibition leads to shorter fibers following the S1 nuclease treatment in KB2P-P2 cells, validating that it results in ssDNA gaps accumulation. In our revised manuscript, we have added the following text in lines 335-339:

“To corroborate that the heightened SSB signaling occurring upon EXO1/FEN1 inhibition originates from ssDNA gap accumulation during DNA replication, we performed a DNA fiber assay with S1 nuclease treatment, as detailed by Quinet et al, (2017). Notably, ssDNA specific cleavage of the replication forks with S1 nuclease led to a significant reduction of the fiber length specifically in LNT1-treated KB2P-P2 cells (Figure 4E). This finding confirms that ssDNA gaps are the predominant source of damage in these cells.”

4. The authors used the LNT1 inhibitor to target both FEN1 and EXO1. Insufficient description of this inhibitor is provided, including whether it has additional targets.

We thank the author for this comment. The inhibitor is described in more detail in the paper that we have cited: Nat Chem Biol 12, 815–821 (2016). The compound has been found to inhibit FEN1 and EXO1 with equal potency. The authors also describe that the inhibitor is ineffective against other proteins with highly similar active sites. This information is now provided in lines 287-290 of the revised version of the manuscript:

“ To elucidate this, we utilized the EXO1/FEN1 inhibitor LNT1 , previously demonstrated by (Exell et al, 2016) to potently inhibit both FEN1 and EXO1 with equal potency, while showing no inhibitory effect on other proteins with conserved active sites.”

5. Regarding to the PARP trapping assay and PARylation immunoblot, the authors may need to deplete FEN1 or EXO1 individually to confirm their separate contributions. Additionally, in Figure EV 3D, LNT1 treatment seems not to induce pRPA2 S4/8 or γ H2AX (Lane 4 vs 8).

The purpose of the PARP trapping assay and PARylation immunoblot in this setting was to test whether the FEN1/EXO1 inhibitor, as a compound, is contributing to PARP1 trapping. Although repeating this assay following independent depletion of FEN1 and EXO1, i.e., with siRNA pools, will answer a different question, we performed this experiment, and the results are presented in the figure below. Chromatin extracts of the KB2P and KB2P-P2 cell lines were isolated, following the siRNA-mediated depletion of Exo1 and Fen1, and treatment with

or without 500nM of olaparib. Immunoblot analysis of the chromatin-associated PARP1 levels showed that Exo1 and Fen1 depletion do not lead to increased PARP1 trapping, but rather leads to a decrease in the levels of chromatin-associated PARP1, same as observed with EXO1/FEN1 inhibition in Figure EV3B.

We also thank the reviewer for the comment regarding Figure EV3D. The fact that we do not observe an increase in pRPA2 S4/8 or γ H2AX levels upon LNT1 treatment in this setting may be explained by the fact that Western blotting was performed with whole cell lysates that are not S-phase specific and it was carried out after a short incubation time of only 15min with the FEN1/EXO1 inhibitor. S-phase-associated γ H2AX levels, following treatment with the EXO1/FEN1 inhibitor have now been measured (as addressed in comment 2) and they are now presented in Figure 4F.

6. Figure 5C, EXO1 and FEN1 depletion showed a synergistic effect on cell viability in KB2P-P2 cells. In this context, the authors should explore the potential distinct functions between them.

We agree with the reviewer that exploring the potential distinct functions of FEN1 and EXO1 would be of great interest. However, dissecting their different mechanistic roles for the survival of PARG;BRCA2-deficient cells is a complex task. FEN1 and EXO1 are involved in various DNA repair pathways with several overlaps. To define their distinct functions, we think that one would first have to chart the essentialome of FEN1- versus EXO1-depleted cells in PARG-proficient cells. Differences between these can then be tested as the potential cause of the synergistic effect in PARG-deficient cells. We think that this is an interesting question for a follow-up study, and we believe that it is beyond the scope of our current manuscript. We have now added the following sentence in our discussion in lines 464-468:

“Since we do not see complete epistasis of Exo1 and Fen1 knockdown on cell survival, we cannot exclude that EXO1 and FEN1 protect Parg^{-/-} cells by independent, and yet-to-be-determined mechanisms. In general, it would be interesting to define the distinct roles of EXO1 versus FEN1 and test which of these contribute to the increased sensitivity of Parg^{-/-} cells. It is also possible that there is redundancy in their function, but the full activity of both enzymes is required to deal with the lesions induced by the PARG deficiency.”

Minor points:

1. According to the sanger sequencing, both KB2P-P1 and KB2P-P2 show large deletions (Figure EV1A). The authors should include a larger sequence region to show the whole deletion region and the flanking sequence. In Figure EV2E, the author should indicate the

stop codon of the frameshift mutation and if any essential domain of PARG is damaged by the mutation.

We thank the reviewer for this comment. The length of the deletions in KB2P-P1 and KB2P-P2 clones are 624bp and 33312bp, respectively. Therefore, it is rather complicated to include the whole alignment in the manuscript. We now present in the new Figure EV1A the alignment fragments including both the sgRNA sequence and the flanking sequence regions. We believe that this is a sufficient representation of the extensive deletions in our clones.

The stop codon in the KP-P1 sequence is now indicated in Figure EV2F.

2. The authors should indicate the treatment of Olaparib and MMC in figure EV1C.

We have annotated 0.01% MMS in Fig. EV1C. Olaparib is already indicated in EV1B.

Referee #2:

In the manuscript provided, authors present work showing that EXO1 or FEN1 inhibition causes synthetic lethality in cells with both BRCA2 and PARG loss. The same authors previously showed that loss of PARG causes PARP inhibitor resistance in BRCA2 mutant mice - this makes the case for identifying the weaknesses in tumor cells caused by both BRCA2 and PARG interesting (although, as the authors acknowledge, PARG defects have not been seen in patients with PARP inhibitor resistance).

The data is well presented and experiments appear to be well carried out and analyzed. The flow of the manuscript is clear and overall the manuscript makes a clear contribution to the literature and understanding of the subject area (of that there is no doubt).

We thank the reviewer for his/her positive feedback.

My only suggestions are:

1. The analysis is very focused on mouse cells, with only a small number of experiments in human cells with a BRCA2 mutation (DLD1, which has an engineered BRCA2 mutation, not a naturally occurring one). I could not see any validation of the key synthetic lethal phenotypes in human tumor cells with endogenous BRCA2 LoF mutations such as Capan 1 and PEO1. There are also HR proficient derivatives of these that could also be used to prove the phenotypes seen are dependent or independent of BRCA2 status as well. These experiments are important to eliminate the possibility that these are mainly mouse phenotypes. This is also important as there looks to be some inconsistency between what is reported in human tumor cells (see below) and what happens in engineered mouse cells.

We agree with the reviewer that it is useful to validate our findings in human tumor cells with endogenous BRCA2 mutations. For this purpose, we used the BRCA2-deficient PEO1 and the paired BRCA2-reconstituted PEO1 C4-2 cell lines. Our experimental approach and results are described and displayed below:

Following the lentiviral transduction of one Scrambl control and two Parg shRNA hairpins in the PEO-1 and PEO-1 C4-2 cell lines, we confirmed the efficient depletion of PARG in both cell lines by Western blot analysis (A). In order to first test whether the BRCA2-deficient PEO1

lines respond to PARPi treatment and whether PARG depletion leads to PARPi resistance, we performed a long-term viability assay in the presence of increasing concentrations of olaparib. Both PEO-1 and PEO-1 shScr lines exhibited mild responses to nanomolar olaparib concentrations that are clinically relevant (B). In contrast our previously used mouse KB2P and human DLD BRCA2^{-/-} lines had exhibited a much stronger response to PARP inhibition, already with 50nM treatment (C). Due to the partially PARPi resistant phenotype of the WT and shScr PEO-1 lines, the additional depletion of PARG conferred only a slight increase in their viability upon olaparib treatment (B). Our results are consistent with the literature, showing that the patient-derived tumor cell lines PEO-1 and Capan-1 require much higher concentrations of PARPi to achieve a strong sensitization effect in culture, with IC₅₀ values around 1 μ M^{1,2}. We suspect that this may be a result of hypomorphic mutations or secondary adaptations.

Nevertheless, when we tested the generated cell lines in long-term viability assays with increasing concentrations of the EXO1/FEN1 inhibitor LNT1, we confirmed that the BRCA2-deficient PEO-1 cells are more sensitive to LNT1 treatment compared to the BRCA2-proficient PEO-1 C4-2 line (D). Moreover, this effect is exacerbated upon additional depletion of PARG. Again, the BRCA2;PARG-deficient PEO-1 lines required higher LNT1 concentrations, compared to the BRCA2;PARG-deficient KB2P and DLD-1 cell lines (D).

Although our new results confirm the sensitization effect of EXO1/FEN1 inhibition in these human tumor-derived cell lines, the use of the PEO-1 cell line does not serve as an equally potent HR-deficient or PARPi-resistant tumor model. We believe that this is also true for the Capan-1 cells, which shows an even milder response to olaparib treatment according to the literature². We therefore think that the engineered mouse KB2P and human DLD BRCA2^{-/-} cell lines are more efficient models for answering our research question and that the newly generated data are not essential for strengthening the revised version of our manuscript. We suggest to leave these additional data with the PEO-1 cell line out, but if the editor prefers to include them as supplementary data, we are happy to do that.

1. Guillemette, S. *et al.* Resistance to therapy in BRCA2 mutant cells due to loss of the nucleosome remodeling factor CHD4. *Genes Dev.* 29, 489–494 (2015).
2. Tacconi, E. M. C. *et al.* Chlorambucil targets BRCA1/2-deficient tumours and counteracts PARP inhibitor resistance. *EMBO Mol. Med.* 11, e9982 (2019).

2. Is there a reason why Mengwasser *et al.*, 2019, *Molecular Cell* 73, 885-899 (Elledge) is not mentioned or taken into consideration? This manuscript shows rather convincingly that FEN1 is synthetic lethal with BRCA2 in human cell lines. How can the authors explain the observation in Figure 5B where it is only the combined loss of BRCA2 and PARG that causes sensitivity to FEN1 siRNA (with no sensitivity when only BRCA2 and p53 are defective)? Is this a mouse specific phenotype and not seen in human cells (there is no human expt using FEN1 siRNA only using a not very potent FEN1/EXO1 inhibitor in Figure 5D).

We thank the reviewer for this comment. We simply missed adding this publication, and it is now mentioned and cited in the discussion section of the revised manuscript in lines 475-480 as follows:

*“ Furthermore, Mengwasser *et al.* (2019) described a synthetic lethal interaction between FEN1 and BRCA2 deficiency. Although our experiments did not show marked sensitization of BRCA2-deficient cells to FEN1 depletion in the presence of active PARG, it is plausible that HR-deficient cells generally exhibit a strong reliance on EXO1 and FEN1-mediated DSB repair pathways, particularly when their ssDNA repair mechanisms are compromised.”*

Consistent with this manuscript, we also observe a mild sensitization of the DLD BRCA2^{-/-} and the PEO-1 cell lines to FEN1/EXO1 inhibition. However, in our experiments the effect is enhanced upon PARG loss. This observation is valid for both the mouse and human cell lines, making a mouse-specific phenotype very unlikely.

3. It is more common to show dose response curves for the small molecule expts (throughout). In general, the experiments using the low potency FEN1/EXO1 inhibitor are not as convincing as those where genetic inhibition is used.

It is correct that we have used the FEN1/EXO1 inhibitor for several experiments. It has the advantage that the effect is more immediate and there may be a clinical compound derived from it in the future. We are not aware that there is a better FEN1/EXO1 inhibitor available, which is more potent and specific. It is not unusual that chemical inhibition of a protein is not equally effective as the genetic depletion. Since our viability assay observations are consistent between the genetic and chemical depletion of FEN1/EXO1, we are confident that the experiments with the FEN1/EXO1 inhibitor are valid.

Referee #3:

In the manuscript entitled 'PARG-deficient tumor cells have an increased dependence on EXO/FEN1-mediated DNA repair' Andronikou et al. performed two CRISPR-Cas9 whole genome drop-out screen to identify genes essential in the survival of PARG;BRCA2;p53-deficient mice tumour cells using both genetic depletion of PARG and chemical PARG inhibition. Their screens identified various genes essential for the survival of PARG;BRCA2;p53-deficient cells. In particular, they revealed that loss of EXO1 and FEN1 is synthetically lethal with loss of PARG activity in these BRCA2;p53-deficient cells. Furthermore, they demonstrate that PARG;BRCA2-deficient cells exhibit increased replication stress and PAR levels which is further exacerbated upon EXO1 and FEN1 inhibition, resulting in cell death. Lastly, they show that EXO1/FEN1 inhibition (using LNT1 inhibitor) was also toxic to human PARG deficient tumor cells, thus identifying the two enzymes as potential drug target for tackling PARPi resistant tumors with PARG downregulation.

Overall, we found the experiments to be well designed and executed. The findings described here have important implications for future development of PARG inhibitors and we believe that the manuscript should be published in the EMBO journal after minor revisions.

We thank the reviewer for his/her positive feedback.

We just have a few clarifications to suggest.

Main comments:

1. PARG itself was identified as a hit in the CRISPR drop-out screen performed with chemical PARG inhibition. The authors also show that PARG;BRCA2-deficient cells are sensitive to both PDDX-001 and PDDX-004. This is surprising and indicates that both PARG inhibitors

have off-target effects upon PARG loss, as stated by the authors. Could the authors comment further on what these off-target effects could possibly be and what implications do these observations have for the future clinical use of PARGi?

We thank the reviewer for pointing this out. This point was also raised by referee 1 (his/her comment 1). As outlined in our response to referee 1, we have now performed experiments regarding the off-target effect which are presented in Figures EV2A-E.

2. In Figure 4D, the nucleus size appear significantly smaller in the following conditions compared to rest of the images : LNT1+EME treatment in KBP2-NT and KBP2-P2, and EME treatment in KB2P-P2. Could the authors double check that the same magnification was used for all the representative images chosen for the figure? If so, is this smaller nucleus a representative feature of the specific condition tested or an artefact of the area chosen for imaging? If the authors think this is representative of the whole well, could they comment on why they think these specific conditions lead to a reduction in nucleus size?

We thank the reviewer for detecting this mistake. This is indeed an artefact due to different magnifications used. We apologize for this error and we have corrected it in the revised manuscript.

3. Identifying the substrates of PAR triggered by these unresolved DNA lesions is beyond the scope of this paper. However, could the authors comment on what these substrates might possibly be in the discussion? Do the authors believe that it is primarily PARP1 auto-modification as suggested by their schematic (Fig 5E), if so this should be made clear in the text that refers to Figure 5E.

We thank the referee for this suggestion. Since there is a PAR “smear” on the Western Blot it is likely that there is more than PARP1 auto-modification. However, we believe that the S-phase-associated increase in ADP-ribosylation levels is mainly a result of persisting PARP1 auto-PARYlation, which serves as the signal for unligated Okazaki fragments. We have now described in more detail the schematic of Figure 5E in lines 388-393:

“ These observations are consistent with the model that EXO1/FEN1 inhibition elevates ADP-ribosylation levels in PARG;BRCA2-deficient cells, likely due to persisted PARP1 auto-PARYlation on unligated Okazaki fragments. These fragments, remaining as ssDNA gaps, have the potential to evolve into DSBs upon replication. Such DSBs are tolerated in p53-deficient cells, but become lethal in the context of compromised homology-directed DNA repair due to the concurrent loss of BRCA2 (Figure 5E).”

4. In the 'PAR/EXO1 immunofluorescence' analysis paragraph in the Methods section, the authors describe using the polyclonal anti-EXO1 antibody for an immunofluorescence experiment. However, as far as we could see, no experiment looking at EXO1 signal by IF can be found in the results. Could the authors update the methods section accordingly?

We thank the reviewer for spotting this mistake. We have now deleted this sentence in the revised manuscript.

Additional comments:

1. On page 3, line 61, Sharifi et al. 2013 (10.1038/emboj.2013.51) should be cited at the end of this sentence.

We have now added this citation.

2. On page 5, line 103 and page 18, line 451, the authors could add that a newly developed PARGi has recently entered clinical trials (IDEAYA bioscience), making the results described in this study even more relevant for future clinical applications.

We appreciate this useful suggestion and have now added and discussed this information in the discussion in lines 486-491 as follows:

“Notably, PARG inhibitor compounds are already advancing into Phase 1 clinical trials for HR-deficient solid tumors (<https://prn.to/43Jn6zU>). Our study shows that PARG inhibition using PDDX inhibitors induces toxicity in PARG;BRCA2-deficient cells through heightened ADP-ribosylation levels. This observation, potentially attributable to off-target effects, highlights the critical need for meticulous compound evaluation to ensure the effective and safe implementation of PARG inhibitors in the clinic.”

3. Could the authors provide molecular weights for their immunoblotting figures (EV3B and EV3D)? This is especially important for the PAR blots.

Thanks. We have now provided this in Figures EV3B and EV3D.

4. Representative image for KB2P-P2 in Figure EV1C is blurry in both the PAR and Hoechst channel.

Thank you. We have now corrected this.

5. Could a higher magnification be used for the representative images of Figure EV1G. It is currently really hard to even distinguish individual cells with the magnification chosen.

If possible, we prefer to keep this magnification, because it nicely shows the consistency of the PAR levels across many cells.

6. On page 15, line 381, "in" or equivalent phrasing should be added after "understanding"

Thanks. We have now corrected this.

Prof. Sven Rottenberg
University of Bern
Pathobiology and Bern Center of Precision Medicine
Laenggassstrasse 122
Bern 3012
Switzerland

8th Jan 2024

Re: EMBOJ-2023-114851R
PARG-deficient tumor cells have an increased dependence on EXO1/FEN1-mediated DNA repair

Dear Sven,

A (belated) Happy New Year, and thank you for submitting your revised manuscript to The EMBO Journal. It has now been assessed once more by one of the original referees, in light of whose positive feedback (see below) we shall be happy to publish the paper, once the referee's remaining minor suggestions and the following editorial issues have been incorporated during a final round of minor revision:

- Please double-check all citations in the reference list, as some of them appear to be incomplete (e.g. lacking journal name, volume/page numbers). Also, when citing preprints, please make sure to adhere to the format specified in our Guide to Authors: The citation in the text should be: "(preprint: NAME1 et al, YEAR)"; in the reference list: "Author NAME1, Author NAME2, ... (YEAR) article title. bioRxiv doi: XXX"

- Please rename the Conflict of Interest section into "Disclosure and Competing Interests Statement", in accordance with our updated Guide to Authors (<https://www.embopress.org/competing-interests/>)

- Several issues related to figure legends, data representation and statistical analysis still remain to be corrected:

* in Figure Legend 4F, the number of independent replicates (n) has not been specified

* error bars still have to be removed from Figure panels 3C, 5C, 5D and any other instances where n=2, as no meaningful statistical information can be derived from less than 3 independent replicates.

* please make sure to correct the length unit for scale bars in micrographs from "um" to "µm" throughout.

- Finally, during routine pre-acceptance checks, we noted that several panels showing cellular colonies are displayed twice in Figs 2C and 5B. We realize that they represent the control conditions in both cases, but this nevertheless has to be justified and made clear in both respective figure legends.

I am therefore returning the manuscript to you for a final round of minor revision, to allow you to make these modifications and upload the revised files. Once we will have received them, we should be ready to swiftly proceed with formal acceptance and production of the manuscript.

With best regards,

Hartmut

9) Digital image enhancement is acceptable practice, as long as it accurately represents the original data and conforms to community standards. If a figure has been subjected to significant electronic manipulation, this must be clearly noted in the figure legend and/or the 'Materials and Methods' section. The editors reserve the right to request original versions of figures and the original images that were used to assemble the figure. Finally, we generally encourage uploading of numerical as well as gel/blot image source data; for details see: embopress.org/page/journal/14602075/authorguide#sourcedata

At EMBO Press, we ask authors to provide source data for the main manuscript figures. Our source data coordinator will contact you to discuss which figure panels we would need source data for and will also provide you with helpful tips on how to upload and organize the files.

In the interest of ensuring the conceptual advance provided by the work, we recommend submitting a revision within 3 months (7th Apr 2024). Please discuss the revision progress ahead of this time with the editor if you require more time to complete the revisions. Use the link below to submit your revision:

Link Not Available

Referee #1:

The authors have addressed all reviewer concerns. It is an interesting study and will make an important contribution. On a minor note, the model in Fig 5E should show how FEN1 is involved. This typically involves 5'-flap removal. However 5E doesn't show any specific involvement of FEN1 vs. Exo1. This should be addressed to reflect the substrate specificities of each nuclease.

POINT-BY-POINT RESPONSE TO EDITORS' AND REFEREES' COMMENTS

Comments in black and responses in red

Editor:

- Please double-check all citations in the reference list, as some of them appear to be incomplete (e.g. lacking journal name, volume/page numbers). Also, when citing preprints, please make sure to adhere to the format specified in our Guide to Authors: The citation in the text should be: "(preprint: NAME1 et al, YEAR)"; in the reference list: "Author NAME1, Author NAME2, ... (YEAR) article title. bioRxiv doi: XXX"

Thank you. We have now corrected this.

- Please rename the Conflict of Interest section into "Disclosure and Competing Interests Statement", in accordance with our updated Guide to Authors (<https://www.embopress.org/competing-interests>)

Thank you. We have now renamed this section.

- Several issues related to figure legends, data representation and statistical analysis still remain to be corrected:

* in Figure Legend 4F, the number of independent replicates (n) has not been specified

The data are representative of three independent replicates. This is now specified in the legend of Figure 4F.

* error bars still have to be removed from Figure panels 3C, 5C, 5D and any other instances where n=2, as no meaningful statistical information can be derived from less than 3 independent replicates.

Error bars are now removed.

* please make sure to correct the length unit for scale bars in micrographs from "um" to "µm" throughout.

Thanks. This is now corrected throughout the text.

- Finally, during routine pre-acceptance checks, we noted that several panels showing cellular

colonies are displayed twice in Figs 2C and 5B. We realize that they represent the control conditions in both cases, but this nevertheless has to be justified and made clear in both respective figure legends.

Thank you for noting this. This is because the experiments were performed simultaneously and therefore the control conditions were the same. However, in order to avoid any misunderstandings we have now replaced the representative pictures of the control conditions of Figure 5B with new ones, from other biological replicates.

Referee #1:

The authors have addressed all reviewer concerns. It is an interesting study and will make an important contribution. On a minor note, the model in Fig 5E should show how FEN1 is involved. This typically involves 5'-flap removal. However 5E doesn't show any specific involvement of FEN1 vs. Exo1. This should be addressed to reflect the substrate specificities of each nuclease.

We appreciate the referee's support of our manuscript. We have now included in the model of Figure 5E a representation of FEN1/EXO1 involvement in the 5' flap removal during Okazaki fragment processing. The distinct functions of FEN1 and EXO1 during this pathway have not been characterized in mammalian cells and although interesting, this is beyond the scope of our manuscript. However, both FEN1 and EXO1 carry a 5' flap removal activity so they may have a redundant role during this process¹. Therefore our model shows that FEN1/EXO1 inhibitor treatment results in inhibition of both FEN1 and EXO1 5' flap removal activity upon Okazaki fragment processing.

1. *Keijzers, G., Bohr, V. A. & Rasmussen, L. J. Human exonuclease 1 (EXO1) activity characterization and its function on flap structures. Biosci. Rep. 35, (2015).*

Prof. Sven Rottenberg
University of Bern
Pathobiology and Bern Center of Precision Medicine
Laenggassstrasse 122
Bern 3012
Switzerland

23rd Jan 2024

Re: EMBOJ-2023-114851R1
PARG-deficient tumor cells have an increased dependence on EXO1/FEN1-mediated DNA repair

Dear Sven,

Thank you for submitting your final revised manuscript for our consideration. I am pleased to inform you that we have now accepted it for publication in The EMBO Journal.

Yours sincerely,

Hartmut
